

**Modeling SOA contributions of VOC, IVOC and SVOC emissions**
**and large uncertainties associated with OA aging**
Ling Huang[1], Hanqing Liu[1], Greg Yarwood[2,*], Gary Wilson[2], Jun Tao[3], Zhiwei Han[4], Dongsheng
Ji[5], Yangjun Wang[1], Li Li[1*]
[1]School of Environmental and Chemical Engineering, Shanghai University, Shanghai, 200444,
China
[2]Ramboll, Novato, California, 94945, USA
[3]Institute for Environmental and Climate Research, Jinan University, Guangzhou, 510632, China
[4]University of Chinese Academy of Sciences, Beijing, 100049, China
[5]Institute of Atmospheric Physics, Chinese Academy of Sciences, Beijing, 100029, China
*Correspondence to*: Li Li (lily@shu.edu.cn), Greg Yarwood (gyarwood@ramboll.com)
**Abstract**
Secondary organic aerosols (SOA) are an important component of atmospheric fine particulate
matter ($PM_{2.5}$) in China, and elsewhere, with contributions from anthropogenic and biogenic
volatile organic compounds (AVOC and BVOC) and semi- (SVOC) and intermediate volatility
organic compounds (IVOC). Policy makers need to know which SOA precursors are important but
accurate simulation of SOA magnitude and contributions remains uncertain. We reviewed SOA
modelling studies in the past decade that have reported the relative contributions of different
precursors to SOA concentration and the findings have many inconsistencies due to differing
emission inventory methodologies/assumptions, air quality model (AQM) algorithms, and other
aspects of study methodologies. We investigated the role of different AQM SOA algorithms by
applying two commonly used models, CAMx and CMAQ, with consistent emission inventories to
simulate SOA concentrations and contributions for July and November 2018 in China. Both
models have a volatility basis set (VBS) SOA algorithm but with different parameters and
treatments of SOA photochemical aging. BSOA (SOA produced from BVOC) is found to be more
important over southern China whereas SOA generated from anthropogenic precursors is more
prevalent in the North China Plain (NCP), Yangtze River Delta (YRD), Sichuan Basin and Central
China. Both models indicate negligible SOA formation from SVOC emissions as compared to
other precursors. In July when BVOC emissions are abundant, SOA is predominantly contributed
by BSOA (except for NCP), followed by IVOC-SOA (i.e. SOA produced from IVOC) and ASOA
(i.e. SOA produced from anthropogenic VOC). In contrast in November, IVOC becomes the
leading SOA contributor for all selected regions except PRD, illustrating the important
contribution of IVOC emissions to SOA formation. Therefore, future control policies should aim
at reducing IVOC emissions as well as traditional VOC emissions.





While both models generally agree in terms of the spatial distributions and seasonal variations of
different SOA components, CMAQ tends to predict higher BSOA while CAMx generates higher
ASOA concentrations. As a result, CMAQ results suggest that BSOA concentration is always
higher than ASOA in November while CAMx emphasizes the importance of ASOA. Utilizing a
conceptual model, we found that different treatment of SOA aging between the two models is a
major cause of differences in simulated ASOA concentrations. The step-wise SOA aging scheme
implemented in CAMx (based on gas-phase reactions with OH radical and similar to other models)
exhibits a strong enhancement effect on simulated ASOA concentrations and this effect increases
with the ambient OA concentrations. The CMAQ VBS implements a different SOA aging scheme
that represents particle-phase oligomerization and has smaller impacts, or no impact, on total OA.
A brief literature survey shows that different structure and/or parameters of the SOA aging
schemes are being used in current models, which could greatly affect model simulations of OA in
ways that are difficult to anticipate. Our results indicate that large uncertainties still exist in the
simulation of SOA in current air quality models due to the aging schemes as well as uncertainties
of the emission inventory. More sophisticated measurement data and/or chamber experiments are
needed to better characterize SOA aging and constrain model parameterizations.

## 53  1 Introduction

Atmospheric fine particulate matter ($PM_{2.5}$) could reduce visibility (Kampa and Castanas,
2008), affect regional climate directly or indirectly (Liu et al., 2014), and exhibit adverse
impacts on public health (Xie et al., 2014; Zhang et al., 2007). With the substantial emission
reduction of sulfur dioxide ($SO_2$), nitrogen oxides ($NO_x$), and primary particles, the organic
portion of $PM_{2.5}$ becomes increasingly important. Observations show that organic aerosols
(OA) could contribute $30 - 70\%$ of total $PM_{2.5}$ in China, with the secondary portion (i.e.,
secondary organic aerosols, SOA) accounting for a more significant portion than the primary
portion (i.e., primary organic aerosols, POA) (Zhang et al., 2007). SOA is formed via
chemical reactions of anthropogenic and natural precursors (e.g., volatile organic compounds,
VOC) followed by subsequent gas-particle partitioning processes (Murphy et al., 2006). Due
to gained polarity and hygroscopicity during aerosol aging, SOA is associated with stronger
health impacts in cardiorespiratory mortality (Pye et al., 2021).
Meanwhile, numerical air quality models (AQM) are the tool used to evaluate the
effectiveness of emission control policies, quantify regional contributions, and predict
concentrations under future emission scenarios. However, accurate simulation of the
magnitudes and variations of SOA via AQM has always been challenging due to the
complexity of SOA itself and the complicated chemical processes involved in SOA formation.
Numerous modelling studies have attempted to capture the temporal and spatial variations of
observed SOA in China (Li et al., 2019; Li et al., 2020; Lin et al., 2016), with emphasis on





different aspects, including incorporating reactive surface uptake of dicarbonyls and isoprene
epoxides (Chen et al., 2021; Liu et al., 2020), improve hydroxyl radical (OH) simulations by
incorporating nitrous acid sources (Miao et al., 2021), and adding missing SOA precursors
(Wu et al., 2021). The incorporation of semivolatile (SVOC, saturation vapor concentration
($C^*$) between 0.3 μg m$^{-3}$ and 300 μg m$^{-3}$ and intermediate volatility organic compounds
(IVOC, $C^*$ between 300 μg m$^{-3}$ and 3 ×10$^6$ μg m$^{-3}$; together referred to as S/IVOC) emissions
as important SOA precursors has been a vigorously studied topic. Regional or national
emission inventory of S/IVOC emissions were estimated based on either the conventional
emission ratio method (i.e., ratio based on POA, VOC, or naphthalene) or using directly
measured emission factor (Liu et al., 2017; Wu et al., 2021). Chang et al. (2022) proposed a
full-volatility organic emission inventory recognizing that emissions of organic compounds
have a continuous spectrum spanning from low to high volatility. Adding S/IVOC emissions
or the full-volatility organic emissions in AQM has been shown to improve model simulated
SOA concentrations, although uncertainties exist with the estimated S/IVOC emissions (Wu
et al., 2021). For example, (Huang et al., 2021) applied the Comprehensive Air Quality Model
with extensions (CAMx) to model SOA in the Yangtze River Delta (YRD) region, and IVOC
emissions were shown to increase the simulated SOA concentration by 5%-26%. With an
updated national S/IVOC emission inventory, the average deviation between simulated and
observed SOA concentrations in China was reduced by 25% based on the WRF-Chem model
(Wu et al., 2021).
Besides adding the SOA precursors, efforts have also been made to improve the modelling
framework of SOA formation implemented in AQM (Donahue et al., 2009; Robinson et al.,
2007). The two-product model and the volatility basis set (VBS) approach have been
incorporated in commonly used AQM, with the former easy to implement while the latter
better at capturing the SOA chemical aging (An et al., 2022; Lin et al., 2016; Yao et al., 2020).
SOA simulation with the VBS scheme has been shown to generate more reasonable results
than the two-product model (Huang et al., 2021; Lin et al., 2016). However, due to the
complexity of the VBS scheme, quantifying SOA source contribution based on the VBS
approach has been extremely limited and is mainly based on the brute-force method (An et al.,
2022; Chang et al., 2022; Cheng et al., 2009), which is time-consuming and has limitations as
a source apportionment method due to the nonlinearity of secondary pollutants like SOA. The
Particulate matter Source Apportionment (PSAT; Yarwood et al., 2007) scheme implemented
in CAMx (Ramboll 2020), built upon the two-product modelling scheme, can resolve SOA
source attribution within a single model simulation while maintaining the overall model
consistency. A more recent study by Dunker et al. (2019) combines the VBS scheme and the
first-order sensitivity coefficients obtained by the decoupled direct method (DDM) to quantify
source contributions to SOA in Houston, U.S., using the Path-Integral Method (PIM; Dunker,





110    2015).

Although substantial progress has been made in improving SOA simulation, inconsistent
findings were reported among some recent SOA modelling studies in China, which warrants
further investigation. For example, using the GEOS-Chem model, Miao et al. (2021)
simulated much higher wintertime SOA concentrations over eastern China than summer-time
SOA. In contrast, An et al. (2022) and Chang et al. (2022) reported slightly higher SOA in
summer. Both Chang et al. (2022) and Wu et al. (2021) reported higher SOA formed from
IVOC emissions (IVOC-SOA) than SVOC (SVOC-SOA), which is opposite to Miao's (2022)
results showing much higher (by as much as two times) SVOC-SOA than IVOC-SOA in
eastern China. The simulated SOA from conventional anthropogenic VOC (ASOA) also
differs among studies. ASOA is estimated to be negligible (~0.25 μg m$^{-3}$) over the YRD
region, and adding I/SVOC emissions increased SOA concentration by 116%, according to
An et al. (2022). On the contrary, anthropogenic VOC (AVOC) are reported to contribute
most to SOA formation (35.6 – 59.1%) and S/IVOC contributed the least (6.0 – 10.6%) in Li
et al. (2022). These inconsistencies are potentially due to different S/IVOC emission
inventories and different models being applied; thus, no clear conclusions can be drawn.
This study attempts to resolve some of the existing inconsistencies of simulated SOA based
on two widely used air quality models in China, namely CAMx and CMAQ. These two
models, with their abilities to track source attribution, have been frequently employed for air
quality simulations in China. A publicly available national S/IVOC emission inventory
developed for China was used to quantify the SOA contribution from different precursors.
Similarities and differences in the simulated results from the two models are considered with
a focus on contributions of different emission types (POA, VOC, IVOC, SVOC) and the SOA
aging schemes which differ. Although both models are applied using a VBS scheme, the
CAMx 1.5D VBS treats ASOA aging via gas-phase oxidation of the organic gases that exist in
equilibrium with the SOA, whereas the CMAQ VBS treats ASOA aging via oligomerization
of the condensed aerosol. Results from this study illustrate the uncertainties associated with
SOA schemes in AQMs even when the model input data are harmonized. Revealing the
important influence of SOA aging can stimulate future improvements to SOA modeling
schemes.
**2 Methodology**
**2.1 Model configuration**
In this study, two commonly used air quality models - CAMx version 7.10 (Ramboll, 2021)
and CMAQ version 5.3.2 (Appel et al., 2021) were used to simulate SOA concentrations in
China for July (representing summer) and November (representing fall) of the calendar year



2018. The models are applied over the same domain with input data developed from the same
sources, although each model has its own processes for handling input data. The modeling
domain covers the entire China with a spatial resolution of 36 km (Figure 1). The Weather
Research and Forecasting (WRF) model (version 4.0, (Skamarock and Klemp, 2008) was
applied to simulate meteorological fields, and WRF model configurations were summarized
in our previous studies (Huang et al., 2021). Anthropogenic emissions within China utilized
the Multi-resolution Emission Inventory for 2017 (MEIC, http://www.meicmodel.org,
accessed on 25[th] September 2021) developed by Tsinghua University; emissions outside
China are based on the Emissions Database for Global Atmospheric Research (EDGAR,
http://edgar.jrc.ec.europa.eu/index.php, accessed on 25[th] September 2021) for the year 2010.
Specifically, MEIC provides VOC emissions speciated for Carbon Bond 2005 (CB05) and
SAPRC 2007 (SAPRC07), which were combined to generate speciated VOC emissions for
the CB6 mechanism (see details in Supporting Information). Biogenic emissions were
calculated using a recent offline version of the Model of Emissions of Gases and Aerosols
from Nature (MEGAN version 3.2, http://aqrp.ceer.utexas.edu/projects.cfm, accessed on 25[th]
September 2021).
The CAMx model configuration included the CB6 photochemical gas-phase mechanism
(Yarwood et al., 2010), the static two-mode coarse/fine (CF) PM chemistry option with
ISORROPIA inorganic gas-aerosol partitioning scheme (Nenes et al., 1998), the Regional
Acid Deposition Model (RADM) aqueous phase chemistry, the Zhang dry deposition option
(Zhang et al., 2003) and wet deposition. In terms of the SOA modeling scheme, CAMx
provides the option to select one of two schemes: a traditional two-product scheme (named
SOAP) or a 1.5-D VBS scheme which is a simplified version of the 2-D VBS scheme (Koo et
al., 2014). This study used the CAMx 1.5D VBS scheme because it is more similar than
SOAP to the CMAQ VBS SOA scheme. Compared with the SOAP scheme, the 1.5-D VBS
scheme treats POA as being semivolatile and includes multi-stage aging of SOA with both
oxidation and fragmentation occurring to describe the evolution of OA in terms of oxidation
state and volatility. For CMAQ simulations, the model configuration included the CB6
gas-phase mechanism, the AERO7 aerosol scheme (Appel et al., 2021), the RADM aqueous
phase chemistry, and ISORROPIA inorganic particulate thermodynamics. Both CMAQ and
CAMx consider SOA formation from traditional anthropogenic VOC (e.g., benzene, toluene,
and xylene) and biogenic VOC (i.e., isoprene and monoterpenes), but there are several
differences. For example, CMAQ includes SOA formation from isoprene oxidation
production (IEPOX) in the aqueous phase. An important difference between the CAMx and
CMAQ VBS schemes is that in CAMx, each volatility bin can be oxidized continuously to the
next lower volatility bin in a step-wise manner providing a dynamic SOA volatility
distribution, whereas in CMAQ the SOA is not oxidized and so maintains a static volatility



distribution. This point will be discussed more in Section 3.4.

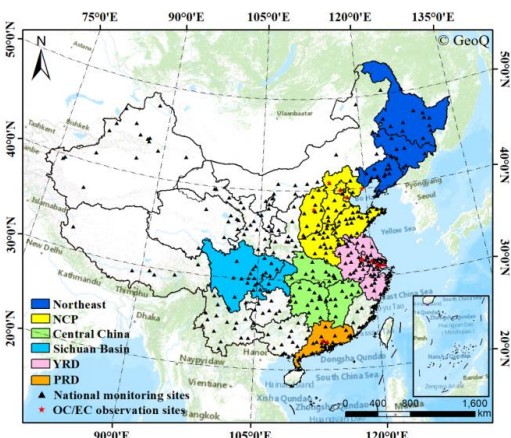

**Figure 1** Modeling domain in this study with definitions of key regions and locations of
national monitoring sites (in black triangles) in 364 major cities and sites with OC/EC
observations (in red star) in 13 cities.
**2.2 S/IVOC emissions**
To evaluate the contributions of S/IVOC emissions to SOA formation, we included a gridded
(0.25 °×0.25 °) monthly S/IVOC emission inventory from industry, residential, transportation,
power plants, and shipping for the year 2016 (Wu et al., 2019). The total reported SVOC and
IVOC emissions were 2,881 and 6,718 Gg, respectively, with industry and residential sectors
being the dominant contributor. For SVOC emissions on an annual scale, industrial and
residential sources accounted for 12.6% and 64.6%, respectively; for IVOC emissions, these
two sectors accounted for 63.1% and 15.5%, respectively. Spatially, S/IVOC emissions are
mainly distributed in China's highly industrialized and urbanized regions, for example, the
Beijing-Tianjin-Hebei (BTH) and YRD regions (Wu et al., 2021). Regarding the monthly
variations, S/IVOC emissions are higher in winter (due to residential sources) and lower in
summer, with winter season (i.e., three months) emissions accounting for 31% of annual total
emissions.
For the CAMx simulations, IVOC were labelled as 'IVOG', 'IVOD', 'IVOB', and 'IVOA',
each representing IVOC emissions from gasoline engines, diesel engines, biomass burning,
and other anthropogenic sources; the corresponding SVOC was renamed as 'POA_GV',
'POA_DV', 'POA_BB' and 'POA_OP' (Ramboll, 2021). In CMAQ simulations, IVOC were
renamed pcVOC (Murphy et al., 2017), and SVOC were allocated into POC and PNCOM
based on existing ratios provided in CMAQ (see Table S2 for details).



**2.3 Simulation scenarios**
In order to evaluate the impact of different models and S/IVOC emissions on simulated SOA
concentration, three sets of parallel simulations (each set with CMAQ and CAMx,
respectively, so a total of six) were conducted for July and November 2018 with identical
meteorological fields (Table 1). In the base scenario, only the conventional anthropogenic
emissions and biogenic emissions were included. In the other two scenarios, IVOC emissions
or S/IVOC emissions were included, respectively. The contribution of IVOC and SVOC
emissions to SOA was quantified by subtracting base case results (e.g., CAMx_IVOC minus
CAMx_base to get IVOC contribution, etc.). Different OA components, including POA,
anthropogenic SOA (ASOA), and biogenic SOA (BSOA), were distinguished in each scenario
because they are resolved by the model OA schemes.
**Table 1** Model scenarios

| Scenario | AQM | Anthropogenic VOC emissions | IVOC emissions | SVOC emissions |
|---|---|---|---|---|
| CAMx_base | CAMx v7.10 | MEIC | none | none |
| CAMx_IVOC | | MEIC | Wu et al. (2021) | none |
| CAMx_S/IVOC | | MEIC | Wu et al. (2021) | Wu et al. (2021) |
| CMAQ_base | CMAQ v5.3.2 | MEIC | none | none |
| CMAQ_IVOC | | MEIC | Wu et al. (2021) | none |
| CMAQ_S/IVOC | | MEIC | Wu et al. (2021) | Wu et al. (2021) |


**2.4 PM$_{2.5}$ and OC/EC observations**
Simulated concentrations of PM$_{2.5}$ and SOA were compared against surface observations of
hourly PM$_{2.5}$ at 364 national monitoring sites and organic carbon (OC)/elemental carbon (EC)
were compared at a limited number of sites. Observations of PM$_{2.5}$ were obtained from the
China National Environmental Monitoring Centre (http://www.cnemc.cn/, accessed on 25[th]
September 2021). Hourly observed OC/EC during July and November 2018 at 13 monitoring
sites (see detailed locations and observation periods in Table S3) were used to evaluate
simulated SOA by applying the minimum OC/EC ratio method (Cao et al., 2004; Castro et al.,
1999) to the observations. The OC/EC sites were mainly located in North China, East China,
and South China. An OA/OC ratio of 1.6 is used to convert SOC/OC to SOA/OA for direct
model comparison (Feng et al., 2009). Model performance was evaluated based on commonly
used statistical metrics, including the mean bias (MB), normalized mean bias (NMB),
normalized mean error (NME), and FAC2 (see Table S4 for definitions).



## 3 Results and discussions

### 3.1 Brief review of existing SOA simulation studies in China

Table 2 summarizes existing SOA simulation studies conducted using different AQMs in China during the past ten years. Earlier studies used two-product SOA schemes and generally underestimated SOA. For example, based on the two-product scheme implemented in WRF-Chem, Jiang et al. (2012) found a maximum underestimation of 75% in simulated SOA during 2016 over China. In another study by Li et al. (2017), although simulated SOA concentration increased nearly four times after increasing aromatic emissions and changing SOA yields, the model still underestimated observed SOA by 72% on average. To better represent the physical and chemical attributes of SOA in AQMs, Donahue et al. (2006) introduced the VBS scheme, which considers gas-aerosol partitioning and chemical aging of both POA and SOA. Compared to the two-product approach, VBS is shown to improve model simulations of SOA substantially. Han et al. (2016) simulated SOA in eastern China and found that the VBS approach could well reproduce the observed SOA at four monitoring sites both in magnitudes (2.8 μgC/m$^3$ vs. observed value of 3.3 μgC/m$^3$) and in terms of the SOA/OA ratio (33% vs. observed value of 32%). Lin et al. (2016) applied the VBS approach in CAMx to simulate SOA concentration and source attribution in Beijing urban area in the summer; results show that the VBS approach substantially improved simulated SOA concentration (from 0.73 μg/m$^3$ to 5.3 μg/m$^3$ at a station in Beijing); however, the model still under-predicted observed SOA values with NMB of -72%. Besides improving the SOA modeling scheme, the inclusion of S/IVOC emissions (in addition to VOC) as missing SOA precursors was also reported with improved performance of SOA simulations. Based on a national S/IVOC emission inventory developed for six sectors (i.e., industry, residential, transportation, power plants, shipping, and biomass burning), simulated SOA concentration increased by 122% in the PRD region and the fraction of observed SOA resolved by model increased from 18% to 40%. In YRD, Huang et al. (2021) found that the simulated SOA concentration could increase by up to 75.6% with the addition of IVOC emissions under the VBS scheme, but simulated OA was 38.5% lower than observations. In a most recent study by Chang et al. (2022), a full-volatility organic emission inventory developed for China was shown to better reproduce observed OA concentrations as well as the split between POA and SOA. Other efforts to improve SOA simulation include adding SOA formation from reactive surface uptake of dicarbonyls and IEPOX/MAE (Hu et al., 2017), updating SOA yields from S/IVOC, considering additional nitrous acid sources to enhance precursor oxidation rates (Miao et al., 2021), incorporation of $O_3^-$ and $NO_3^-$ initiated SOA formation pathways (Wu et al., 2021).



As shown by Table S5, various S/IVOC emission inventories have been developed for
different sources and regions, yet large uncertainties exist. For example, Liu et al. (2017)
obtained a national total of 200.4 Gg IVOC from mobile sources for the year 2015 based on
the emission factor method, which is slightly lower than the value (241.2 Gg) reported by
Wang et al. (2022) and much higher than Wu's result (134.4 Gg; Wu et al. 2019) based on the
IVOC/POA ratio. Using three different estimation methods, Miao et al. (2017) reported a
range of 3.8 – 6.6 Tg anthropogenic IVOC emissions for 2014 in China, which is 30% – 60%
lower than the value (9.6 Tg) reported by Wu et al. (2021) for 2016. Regionally, Huang et al.
(2021) calculated the IVOC emissions in the YRD region, and IVOC emissions could be
different by a factor of 6.4 – 69.5 when different methods were applied. The relative
contributions from different sources also differ among studies, further increasing the
uncertainties of SOA source tracking. Chang et al. (2022) indicate that domestic combustion
(including fossil fuel and biomass) and open biomass burning represents the dominant
contributors to S/IVOC emissions. In contrast, industry and residential sources contribute 78%
of total S/IVOC emissions, according to Wu et al. (2021). These results indicate significant
uncertainties associated with S/IVOC emission inventory and warrant more measurement data
to better constrain the emission estimation in the future. In this study, we utilized the emission
inventory developed by Wu et al. (2021) and focused on the similarities and differences
between the models while temporarily ignoring the uncertainties associated with the S/IVOC
emission inventory itself.





**Table 2** Summary of findings from SOA simulation studies in China

| Reference | Model | Year/Season | Region | SOA modeling scheme | S/IVOC emissions | Summary of findings |
|---|---|---|---|---|---|---|
| Fu et al. 2011 | GEOS-Chem | 2006/all four seasons | China | two-product | No | Secondary formation accounts for 21 % of Chinese annual mean surface OC in the model |
| Jiang et al. 2012 | WRF-Chem | 2006/all four seasons | China | two-product | No | The model well captured spatial and temporal characteristics of OC and EC, but the simulated SOA concentration was underestimated by up to 75%. |
| Li et al. 2017a | RAMS-CMAQ | 2014/fall | China | two-product | No | By increasing aromatic emissions and modifying model parameters, simulated SOA concentration increased by nearly four but still underestimated the observed SOA by an average of 72%. |
| Lin et al. 2016 | WRF-CAMx | 2007/summer | BTH | VBS | No | VBS approach substantially improved hourly, daily, and monthly SOA simulations but still largely underestimated observed SOA with a normalized mean bias of -72%. |
| Han et al. 2016 | RAQMS[a] | 2009/spring | East China | VBS | No | Adopting the VBS approach with chemical aging suggests ASOA is the dominant SOA component over east China in springtime. |
| Zhao et al. 2016a | WRF-CMAQ | 2010/all four seasons | China | VBS | Yes | OA aging and IVOC emissions increase OA and SOA concentrations in China by about 40%. |
| Li et al. 2016 | RAMS-CMAQ | 2014/winter | China | VBS | No | Updated to incorporate SOA production from isoprene and sesquiterpenes and to account for the SOA production rate dependence on NOx and SOA aging. Modeled monthly mean SOA concentrations were high in central and eastern China and low in western regions. |
| Hu et al. 2017 | WRF-CMAQ | 2013/all four seasons | China | VBS | No | The model included the treatment of isoprene gas-phase chemistry that leads to the production of IEPOX and MAE. It also includes an updated SOA mechanism with updated VBS and SOA formation from reactive surface uptake of dicarbonyls, IEPOX, and MAE. |
| Wu et al. 2019 | WRF-Chem | 2008/winter | PRD | VBS | Yes | The simulated SOA increased by 161% with the input of S/IVOC emissions over the PRD region. |
| Li et al. 2020 | WRF-NAQPMS | 2014/winter | China | VBS | Yes | After adding S/IVOC emissions, the maximum concentration of SOA reached up to 50 μg/m³ in Beijing and Shijiazhuang. SOA/OA ratio is around 50% in most areas of the BTH region. |
| Liu et al. 2020 | WRF-CMAQ | 2015/winter 2016/winter | YRD | VBS | No | Based on the updated SAPRC-11 mechanism, observed and predicted SOA concentrations were 6.4 μg m⁻³ and 6.9 μg m⁻³ in the winter of 2015 and 5.7 μg m⁻³ and 9.6 μg m⁻³ in the winter of 2016. |
| Li et al. 2020 | CMAQ | 2013/winter, summer | Eastern China | two-product | No | The impact of water vapor partitioning and nonideality of the organic-water mixture on SOA formation is considered in the model. The modified model well captured observed diurnal OA variations in winter but didn't capture peak values under polluted days with a mean fractional bias of -20%. |
| Wu et al. 2021 | WRF-Chem | 2017/winter | China | VBS | Yes | With the addition of S/IVOC emissions, the fraction of observed SOA resolved by the model increased from 18% to 40%. |
| Huang et al. 2021 | WRF-CAMx | 2018/summer | YRD | VBS/two-product | Yes | With 1.5 D-VBS and IVOC emissions, simulated SOA concentration increased by 61% in the YRD region, and SOA/OA ratio matched well with observed values. |
| Miao et al. 2021 | GEOS-Chem | 2014/summer, winter | China | VBS | Yes | With updated emissions, volatility distributions, and SOA yields of SVOC and IVOC and the addition of HONO sources, anthropogenic SVOC and IVOC are shown to be the dominant source of SOA, with a contribution of over 50% over most of China. |
| An et al. 2022 | WRF-CMAQ | 2019/spring, summer 2018/autumn, winter | YRD | VBS | Yes | After adding the S/IVOC emissions inventory, the simulated SOA values in the YRD region increased by 116%. Industry is found to contribute the most to SOA concentration in YRD. |
| Chang et al. 2022 | CMAQ | 2017/winter, summer | China | 2D-VBS | Yes | A full-volatility organic emission inventory was developed and was shown to better reproduce observed OA concentration and SOA/OA ratio. Volatile chemical products (VCPs), domestic combustion, and biomass open burning are three leading sources of SOA. |
| Li et al. 2022 | WRF-RAQMS | 2018/summer | Eastern China | VBS | Yes | After adding to the S/IVOC emissions inventory, SOA formation from S/IVOC emissions was lower in summer than in winter. Enhanced contribution to SOA from aqueous uptake and reaction of GLY and MGLY in summer than in winter were modeled. |







**3.2 Model performance evaluation**

Figure 2 shows the spatial distribution of monthly averaged $PM_{2.5}$ concentration simulated under CAMx_S/IVOC and CMAQ_S/IVOC scenarios. Overall, the two models are able to capture the spatial distributions and seasonal variations of observed $PM_{2.5}$ with MB of -4.1 – 8.3 $\mu g\ m^{-3}$, NMB of -17.1 – 17.5%, and NME of 37 – 49%. Values of NMB and NME meet the criteria standards (NMB within 20% and NME within 45%, except for CAMx NME) proposed by Huang et al. (2021), indicating acceptable model performance for $PM_{2.5}$. Regionally, simulations over NCP and Sichuan Basin show an overestimation with MB of -8.2 – 24.8 $\mu g\ m^{-3}$ and NMB of -16.0 – 59.6% (Table S6). CAMx generally tends to simulate higher $PM_{2.5}$ concentrations (by 13.4 – 60.2%) than CMAQ. We further evaluated simulated SOA at a limited number of observation sites (Figure S1 and Figure S2), noting that the "measured" SOA is diagnosed from measurements of total OC as discussed above. Model performance for simulated SOA varies across different sites, with a general pattern of overestimation in July and underestimation in November (Table S7).

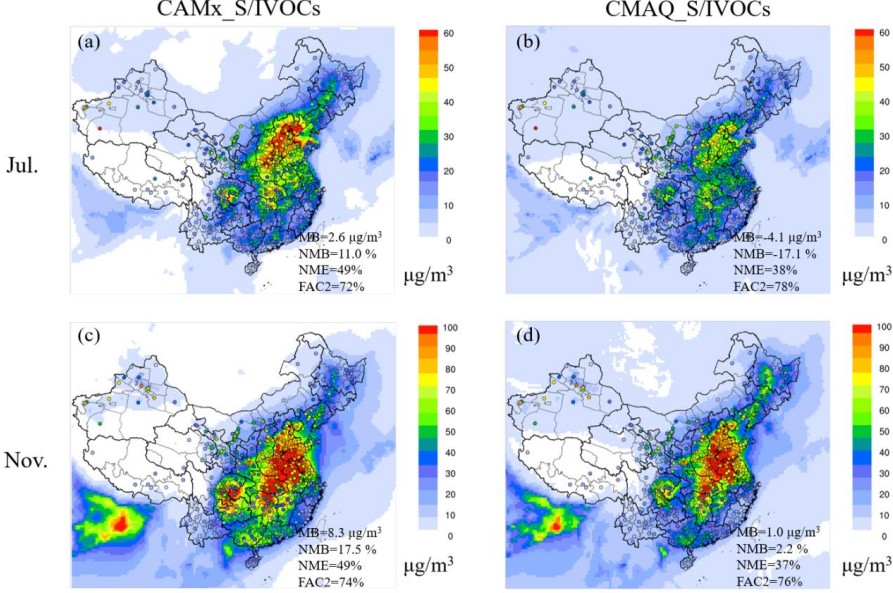

**Figure 2** Spatial distributions of simulated and observed (filled circles) $PM_{2.5}$ concentration (μg/m) in July (upper row) and November (bottom row) 2018 based on CAMx (left column) and CMAQ (right column) simulations including S/IVOC emissions.

**3.3 OA components simulated by CAMx and CMAQ**

Figure 3 shows the spatial distribution of different OA components simulated by CAMx and CMAQ for July and November. Five OA components, including POA, ASOA, IVOC-SOA, SVOC-SOA, and BSOA, were identified either as direct model outputs (POA, and BSOA) or



obtained as the differences between different modeling scenarios listed in Table 1. Table S8
shows the domain averaged concentration for each OA component by selected regions.

*Primary organic aerosols (POA)*

POA represents organic aerosols that are directly emitted from anthropogenic sources (e.g.,
residential wood combustion). In the VBS scheme, POA is allowed to vaporize into gaseous
phase and undergo further oxidation processes to form SOA. The magnitudes and spatial
distributions of POA agree well between CAMx and CMAQ results as well as previous
studies (e.g., Hu et al., 2017), with high concentrations simulated over Northeast China, NCP,
northern YRD, and Sichuan Basin. Seasonally, simulated POA concentration in November is
much higher than July as a result of higher emissions. In July, POA ranged from 0.3 μg m$^{-3}$
(CMAQ PRD) to ~1.9 μg/m$^3$ (CAMx Sichuan Basin) and accounts for less than 10% of the
total OA concentrations for all selected regions, except Sichuan Basin (~15%), suggesting a
relatively minor role compared to SOA during the summer season. In November,
domain-averaged POA ranged from 1.3 μg/m$^3$ (CMAQ PRD) to ~ 6.7 μg/m$^3$ (CMAQ
Northeast) and became the dominant OA component in Northeast China (~70%). For other
regions in November, POA contributed from ~10% (in PRD) to ~40% (in NCP) of the total
OA concentration, suggesting that SOA is still the dominant OA component for most regions.

*Secondary organic aerosols (SOA)*

Except for Northeast China in November, SOA concentration exceeds that of POA and
different SOA components show distinctly different seasonal and spatial variations. Biogenic
SOA (BSOA) is mainly concentrated over the southern part of China where BVOC emissions
are abundant, whereas anthropogenic-oriented SOA (i.e., ASOA and IVOC-SOA) are mainly
concentrated over the northern part of China in July and shifted southward in November.
Seasonally, BSOA concentration is much higher in July, with maximum monthly-averaged
BSOA concentration exceeding 15 μg/m$^3$ over specific areas in Central China. In contrast,
ASOA and IVOC-SOA concentrations are higher in November for most of the regions. Figure
4 and 5 shows the subdomain averaged relative contribution of different OA components for
selected regions. In July, BSOA represented the dominant SOA component (except NCP),
accounting for 45.5% – 60.2% in the Northeast to as much as 71.5% – 82.7% in Central
China. In November when BVOC emissions are dramatically lower, the relative contribution
of BSOA decreases sharply. Nevertheless, BSOA still represents the most abundant SOA
component (by 37.1% – 51.0%) for PRD in November. The two models simulate similar
IVOC-SOA concentrations with a maximum of up to 7.3 μg/m$^3$ in July and 8.7 μg/m$^3$ in
November. The relative contribution of IVOC emissions to SOA ranged from 13.4% – 13.8%
in PRD to 35.0% – 42.5% in NCP for July and 26.3% – 29.7% in PRD to 51.4% – 62.5% in



NCP for November, suggesting an important contribution of IVOC emissions to SOA,
especially during November. Compared to other components, both models suggest that SOA
generated from SVOC emissions (i.e., SVOC-SOA) are negligible (<0.5 μg m$^{-3}$), accounting
for less than 5% of SOA concentration for most regions.
Differences are observed in the magnitude of simulated BSOA and ASOA concentrations
between the two models, although the spatial distributions and seasonal variations are
consistent. On the one hand, CMAQ tends to simulate slightly higher BSOA concentrations
than CAMx. For example, the domain averaged BSOA concentration over NCP in July is 2.8
μg/m$^3$ based on CAMx results, which is 22.2% lower than that of CMAQ. For Central China
where BVOC emissions are abundant, BSOA estimated from CAMx in July (10.8 μg/m$^3$) is
33.1% lower than CMAQ (14.8 μg/m$^3$). In November, the relative differences in BSOA
between the two models become even larger, although the absolute magnitudes are lower
compared to July. The higher BSOA simulated by CMAQ is likely associated with recent
modifications implemented in CMAQ: (1) the yield of organic aerosol from monoterpene
oxidation was increased; and (2) alpha-pinene was treated explicitly in the model and 30% of
the total terpenes emissions were allocated to alpha-pinene (US EPA, 2019). However, we did
not investigate this difference because our study focuses on anthropogenic emissions. On the
other hand, CAMx predicts much higher ASOA concentration than CMAQ, by up to ~4 times
higher in July and ~2 times higher in November. For instance, CAMx simulated ASOA
concentration in NCP is 5.7 μg/m$^3$ for July as opposed to 1.4 μg/m$^3$ by CMAQ. In November,
CAMx simulated ASOA concentration in YRD is 4.1 μg/m$^3$ compared to 2.4 μg/m$^3$ by CMAQ.
This difference in simulated ASOA concentration is discussed in detail in Section 3.4.
Because of these differences, the relative contribution of ASOA and BSOA varies between the
two models. For CAMx simulations, ASOA accounts for 10.9% (Central China) – 42.1%
(NCP) of total SOA concentration in July and 23.6% (Northeast China) – 38.7% (Sichuan
Basin) in November. For CMAQ simulation, the relative SOA contribution of ASOA ranges
from 4.3% (Central China) – 15.0% (NCP) in July and 16.3% (PRD) – 21.8% (YRD) in
November. In November, CMAQ results suggest that BSOA concentration is always higher
than ASOA, while CAMx emphasizes the importance of ASOA more than BSOA.



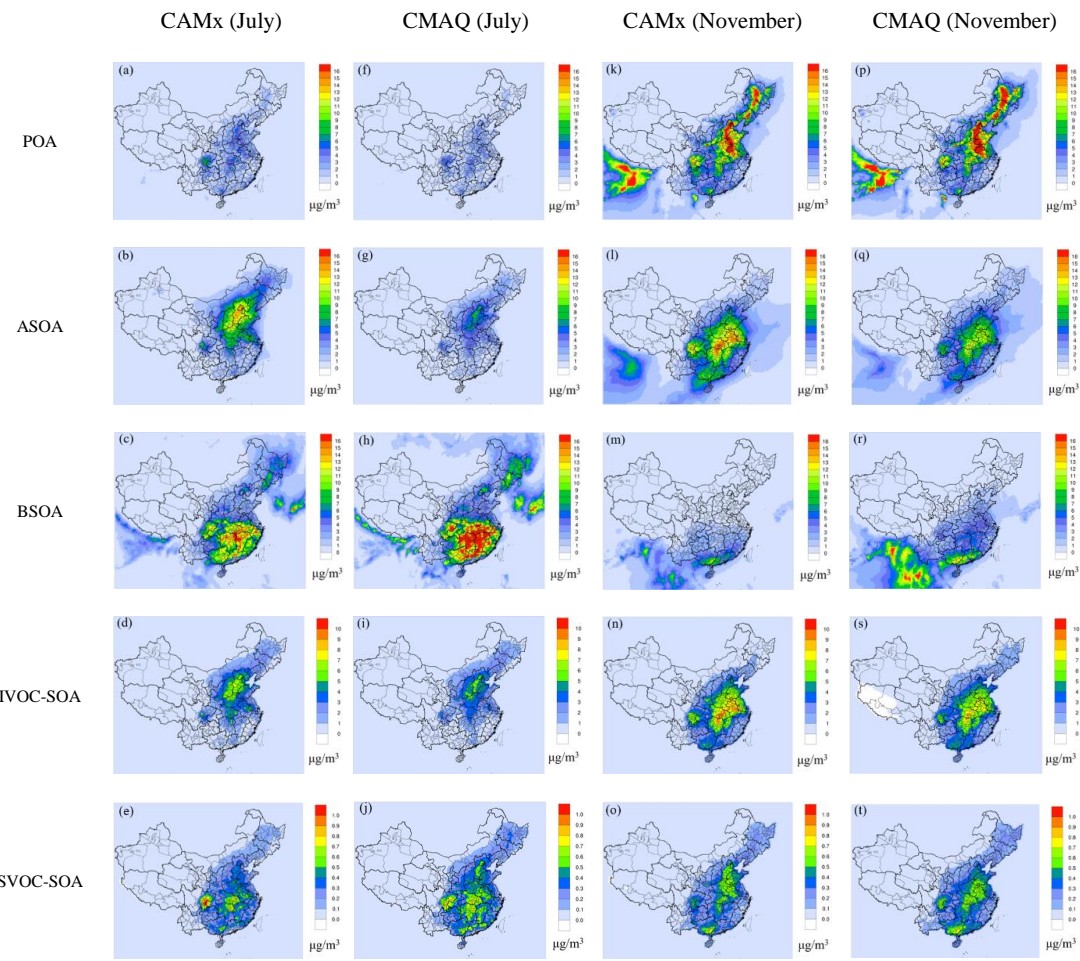

**Figure 3** Spatial distributions of different OA components in July and November




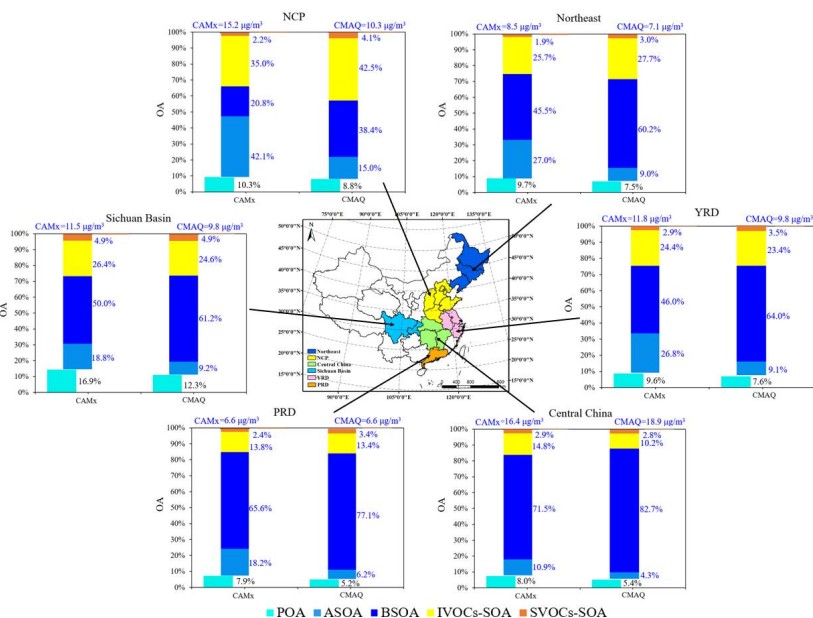

**Figure 4** Domain-averaged OA concentration (labelled on top) and relative contributions of different components (labelled inside Figures) in July 2018 (Note that the relative contributions of ASOA, BSOA, IVOC-SOA, and SVOC-SOA sum to 100%).

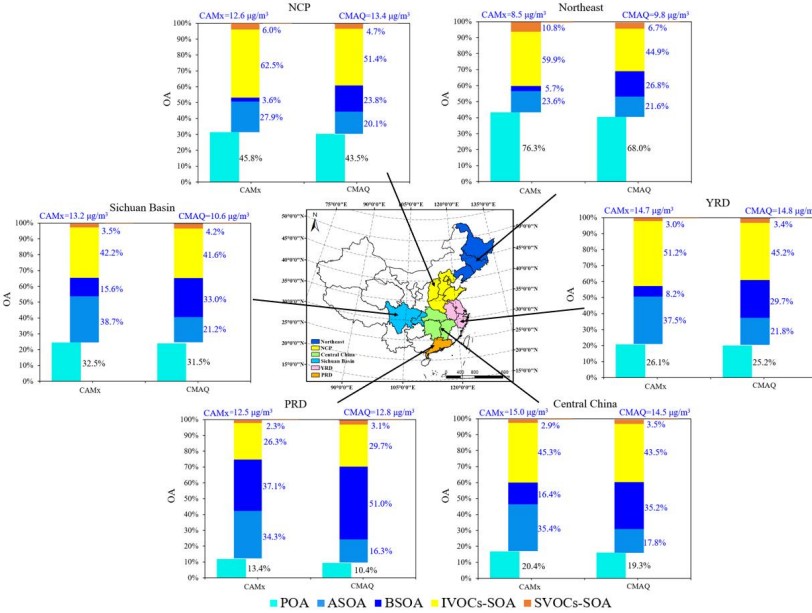

**Figure 5** Domain-averaged OA concentration (labelled on top) and relative contribution of different components (labelled inside Figures) in November 2018 (Note that the relative contribution of ASOA, BSOA, IVOC-SOA, and SVOC-SOA sum to 100%).



**3.4 Accounting for SOA aging**
Although a VBS scheme is implemented in both CAMx and CMAQ, substantial differences
were found in the simulated ASOA concentrations, as shown above, with CAMx predicting
much higher values than CMAQ (Figure 4 and Figure 5). For example, simulated
domain-averaged ASOA concentration in July by CAMx is 5.7 μg/m$^3$ and 2.9 μg/m$^3$ over
NCP and YRD, respectively, but the corresponding values simulated by CMAQ are only 1.4
μg/m$^3$ and 0.8 μg/m$^3$, which are approximately four times lower. The relatively high ASOA
concentrations from CAMx results are consistent with the study by Li et al. (2022), who
applied the regional air quality model system (RAQMS) over east China and reported 48.3%
of SOA was formed from AVOC emissions. Meanwhile, the much lower ASOA
concentrations simulated by CMAQ in July over the YRD region (domain-average: 0.8 μg/m$^3$)
are consistent with another CMAQ study conducted by An et al. (2022), who showed minimal
SOA contribution (<0.5 μg/m$^3$) from AVOC for all four seasons. The significant differences in
simulated ASOA concentrations between the two models are explained by two aspects of the
VBS implementation. First, the SOA molar yields from anthropogenic VOC (i.e., benzene,
toluene, and xylene) are different in the two models, especially under low-NO$x$ conditions
(Table S9). For instance, under the low-NO$x$ conditions, CMAQ SOA is only formed in the
least volatile bin (C$^*$=0.01 μg/m$^3$), whereas CAMx SOA is formed in different bins with
volatility ranging from 1 μg/m$^3$ to 100μg/m$^3$. The total VBS yield (summed over VBS bins)
also differs between the models, as exemplified by the total yield for benzene being 0.146 in
CMAQ vs. 0.596 in CAMx for low-NO$_x$ conditions (Table S9). The total VBS yield does not
translate directly into SOA yield because the effects of bin volatility (C*) and aging must also
be considered, as illustrated below. Nevertheless, the total VBS yields for anthropogenic VOC
are higher in CAMx than CMAQ for all precursors and high/low NO$_x$ conditions. This
difference can be explained, in part, by the influence of wall-losses on chamber experiments
that are used to characterize SOA formation and derive VBS product distributions. The
CAMx 1.5D VBS SOA yields are corrected for vapor wall losses in smog chamber
experiments (Hodzic et al., 2016; Zhang et al., 2014) whereas the CMAQ VBS SOA yields
are not wall-loss corrected (personal communication from Dr. Havala Pye, U.S. EPA).
The second major difference between the two models lies in the treatment of SOA aging. In
the CAMx 1.5D VBS, gas-phase oxidation products in different volatility bins are
continuously oxidized by reactions with OH (with an OH rate constant of 2×10$^{-11}$
cm$^3$/molecule/s) that move mass from higher volatility bins to the next lower volatility bin in
a step-wise manner (for example, from C$^*$=1000 μg/m$^3$ to C$^*$=100 μg/m$^3$ and from C$^*$=100
μg/m$^3$ to C$^*$=10 μg/m$^3$). The CAMx aging scheme will tend to produce more extensive aging
in summer than winter due to higher OH concentrations and greater evaporation of SOA to



the gas-phase in warmer conditions. In contrast, the CMAQ VBS has no gas-phase oxidation
of VBS products and instead converts SOA in VBS bins with $C^* = 1$ μg/m$^3$ to $C^* = 100$ μg/m$^3$
directly to non-volatile SOA at a constant rate of 3.4% per hour. The CMAQ aging scheme,
representing oligomerization, will tend to produce more extensive aging in winter than
summer due to greater condensation of SOA in colder conditions.
We investigated how the different treatments of SOA aging in the CMAQ and CAMx VBS
schemes influence simulated SOA concentrations by performing simple conceptual
calculations outside the models. Based on Pankow's partitioning theory (Pankow, 1994), the
SOA yield α from a specific precursor (e.g., benzene) is calculated using Eq. (1):

$$\alpha = \sum_{i}^{N} \beta_i \left( 1 + \frac{C_i^*}{C_{OA}} \right)^{-1} \qquad \text{Eq. (1)}$$

where $\beta_i$ is the mass yields from each volatility bin (i), which are the values listed in Table S9;
$C_i^*$ is the volatility defined for each bin, and $C_{OA}$ is the total ambient OA concentration. We
first consider a scenario with no aging in CAMx (CAMx_no-aging). Taking benzene as an
example and assuming $C_{OA} = 1$ μg/m$^3$ in Eq. (1), SOA mass yield (dimensionless units) from
benzene under high and low-NO$x$ conditions is 0.009 and 0.029, respectively. With ambient
$C_{OA}$ increased by ten times (i.e., $C_{OA} = 10$ μg/m$^3$), the gas-particle partition equilibrium shifts
more towards the particle phase and the SOA mass yield for benzene increases to 0.056 and
0.105 for high and low-NO$x$, respectively. Thus we obtain the SOA mass yields for benzene
as a function of $C_{OA}$ shown by solid lines in Figure 6 (similar calculations for toluene and
xylene are presented in Figure S3). Similarly, for CMAQ with no aging (CMAQ_no-aging)
and $C_{OA} = 1$ μg/m$^3$, the SOA mass yield from benzene is 0.021 and 0.145 under high-and
low-NO$x$ conditions, respectively. As $C_{OA}$ increased to 10 μg/m$^3$, SOA mass yield remained
unchanged under low-NO$x$ conditions and increased by a factor of 3.2 under high-NO$x$
conditions, as shown by dashed lines in Figure 6. When oligomerization (aging) over 6 hours
is considered in CMAQ (CMAQ_aging), SOA yield for benzene is enhanced by
approximately a factor of 2 under the high-NO$x$ condition, but there is no enhancement under
low-NO$x$ condition because the CMAQ VBS always treats this aerosol as being non-volatile
and therefore unaffected by aging.
The effect of aging can be much stronger in CAMx than in CMAQ, depending upon the
modeled OH radical concentration. We performed a simple offline sensitivity analysis
assuming a representative winter OH concentration of $5 \times 10^5$ molecules/cm$^3$. As shown in
Figure 7, an initial VBS distribution favoring higher volatility bins (i.e., towards bin 5)
evolves over time with aging (i.e., OH-reaction of the gas-phase fraction in each bin) to favor
lower volatility bins (i.e., towards bin 1) which reduces overall volatility and increases aerosol



yield. An OH exposure of $10^{10}$ molecules/cm$^3$, equal to an exposure time of 5.75 hr, was used
so that the aging effects shown for CAMx and CMAQ are for a similar duration. With this
amount of aging/oxidation effect, the VBS bin molar yields for benzene under high-NO$x$
become 0.130, 0.580, 0.400, 0.177, and 0.048 as compared to 0.035, 0.108, 0.185, and 0.268
with no aging (see Table 6). These aged yields are converted to SOA mass yields under
different total OA concentrations using Eq. (1), as shown by the red symbol line in Figure 8.
For a $C_{OA}$ = 1 μg/m$^3$ with ~6 hours of aging, CAMx simulated a SOA mass yield of 0.322
from benzene under the high-NO$x$ condition as opposed to 0.009 without aging, an
enhancement greater than a factor of 30. This aging effect becomes even stronger as the
ambient OA concentration increases. These conceptual calculations reveal significant
differences in the magnitude of SOA enhancement produced by SOA aging applied with the
CAMx and CMAQ VBS schemes.
A literature review reveals that different treatments of SOA aging, or no aging, are widespread
in the application of VBS schemes. For example, Hayes et al. (2015) applied box models with
multi-generating aging parameterization to simulate SOA in Los Angeles during CalNex 2010,
which was shown to over-predict urban SOA at photochemical age larger than 1 day. With
several alternate model configurations, Dzepina et al. (2011) predicted SOA mass in Mexico
City during MILAGRO 2006 and found that a scenario with multi-generational SOA aging
scheme and no S/IVOC emissions could successfully predict observed SOA but adding
S/IVOC emissions resulted in large over-predictions. Jiang et al. (2012) applied WRF-Chem
to simulate SOA in China with no aging scheme and concluded that omitting chemical aging
of SOA and POA might be the main reason for their underestimation of SOA by up to 75%.
Han et al. (2016) contrasted simulated SOA results based on the VBS scheme with and
without considering aging in RAQMS and found that a VBS scheme with aging
under-estimated SOA by only 15%, whereas VBS without aging under-estimated SOA by

483    70%.

The examples of differing OA model assumptions and outcomes found in this brief literature
review and in our simulation results emphasize that model simulations of OA are very
sensitive to whether and how SOA aging is represented. A diverse variety of VBS schemes are
being used in air quality models and the predicted OA concentrations may depend upon
scheme structure and/or parameters in ways that are difficult to anticipate. Comparing SOA
yield curves under idealized conditions (e.g., Figure 6) can be helpful for comparing different
schemes. The range of outcomes produced by current OA schemes illustrates that large
uncertainties remain, as well as uncertainties in emission inventories that should be
considered when interpreting results from OA modeling studies. There is a continuing need
for sophisticated measurement data to provide better constraints.




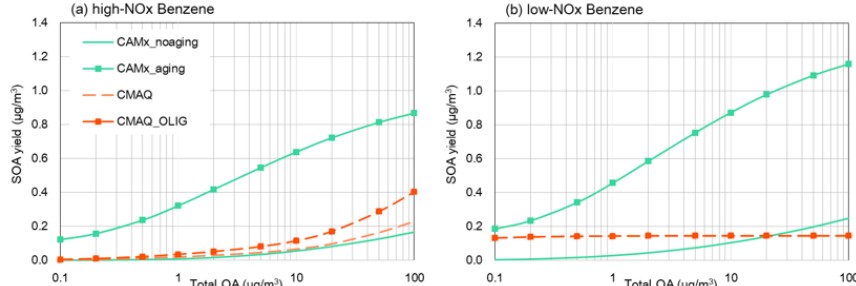

**Figure 6** Conceptual SOA mass yield (μg/m³ SOA formed per μg/m³ precursor reacted) from
benzene under (a) high- and (b) low-NO*x* conditions by different model configurations

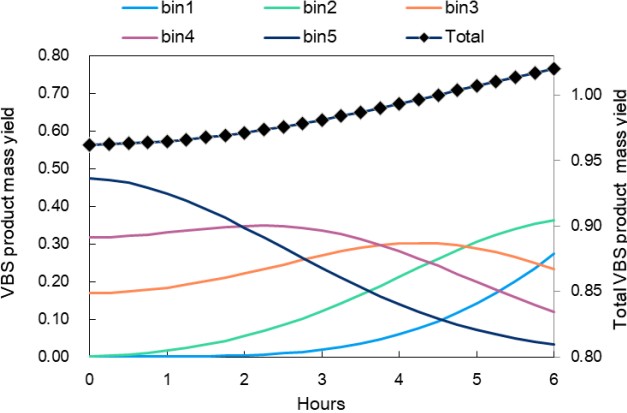

**Figure 7** Aging an initial VBS product distribution that favors higher volatility bins (i.e., near
bin 5) using the CAMx 1.5D VBS scheme with [OH] = $5 \times 10^5$ molecule/cm³ and $k_{OH}$ =
$2 \times 10^{-11}$ cm³/molecule/s changes the volatility distribution to favor lower volatility bins (i.e.,
near bin 1) and increases total VBS product mass yield (due to oxidation) and would increase
aerosol yield (not shown because depends on total OA as illustrated by Figure 8).
**4 Conclusions**
We applied two commonly used air quality models to simulate SOA concentration and
contributions from VOC, IVOC and SVOC emissions over China for July and November
2018, with emphasis on comparing the models and their sensitivity to assumptions within the
VBS schemes. Five OA components, including POA, SOA formed from biogenic VOC
(BSOA), anthropogenic VOC (ASOA), IVOC-SOA, and SVOC-SOA, were resolved. The
POA contribution to total OA is less than 10% in July and ranges from 10% – 40% in
November (except for Northeast China), indicating the dominant role of SOA. Both models
suggest that SOA contributions from SVOC emissions are negligible but IVOC represents an
important SOA precursor, especially during November when IVOC becomes the leading SOA
contributor for all selected regions except PRD (accounting for 26% to 30% in PRD and 45%
to 63% in NCP). Although our modelling results are subject to uncertainties discussed below,



the consistently large contribution of IVOC emissions to total OA indicates that future control
policies should aim at reducing IVOC emissions as well as traditional VOC emissions.
While both models show generally consistent results in terms of the spatial distributions and
seasonal variations of the resolved OA components, differences were found with simulated
ASOA and BSOA concentrations. CMAQ tends to estimate higher BSOA concentration,
while CAMx generates more ASOA. As a result, CMAQ results suggest that BSOA
concentration is always higher than ASOA in November, while CAMx emphasizes the importance
of ASOA. With the help of a conceptual model, we demonstrate that the higher ASOA
simulated by CAMx is attributable to the aging effect on ASOA implemented in CAMx. In
the CAMx 1.5D VBS, SOA formed in higher volatility bins is assumed to undergo further
gas-phase oxidation by OH into lower volatility bins. It is estimated that with 6 hours of aging
at a representative wintertime OH concentration of $5 \times 10^5$ molecules/cm$^3$, SOA aging could
enhance ASOA concentration by an order of magnitude, or more, depending on the total
ambient OA concentration. The CMAQ VBS implements a different SOA aging scheme that
represents particle-phase oligomerization and has smaller impacts on total OA, or no impact,
than the CAMx 1.5D VBS aging scheme. A brief literature survey reveals that a diverse
variety of VBS schemes (aging or no-aging) and/or parameters are being used in air quality
models and could greatly affect model simulations of OA in ways that may be difficult to
anticipate. Results from this study emphasize that improved availability of advanced
monitoring data that resolve OA constituents would improve model evaluation and therefore
model development.
**Financial support.** This study has been supported by the National Natural Science Foundation of
China (Grant NOs. 42005112, 42075144, 41105102) and the Shanghai Sail Program (no.
19YF1415600).
**Competing interests.** The contact author has declared that none of the authors has any competing
interests.

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
