# Peer review of "Modeling SOA contributions of VOC, IVOC and SVOC emissions"

_EGUsphere, 2022_

## Author Comment (AC1)

**Response to reviewers' comments**

Reviewer #1

The manuscript of egusphere-2022-1502 presented the model comparison for OA. Generally, I am impressed by the low presentation quality in this manuscript (especially the introduction section is not well organized) and cannot fully follow the authors' intention. I have judged to reject this manuscript and encourage the resubmission after revisions. I hope the following major and specific comments will help to improve this manuscript.

Response: We thank the reviewer for reading the manuscript carefully and providing helpful comments. We read through all the comments and update the manuscript with improvements suggested by the reviewer's comments. Specifically, we (1) re-organized the introduction section substantially by incorporating the reviewer's suggestions; (2) deleted section 3.1 with points described in the revised introduction. Our point-to-point response is given below.

**Major comments:**

- Introduction: I would like to request a more organized and reader-friendly introduction. Please see my specific comments.

  Response: We have re-organized the introduction part with the following order: the 1st paragraph presents the importance of SOA as before. The 2nd paragraph now describes commonly used SOA formation modules by different air quality models and their advantages and limitations. The 3rd paragraph discusses the importance of S/IVOC emissions as SOA precursors (which was described in Section 3.1). The 4th paragraph describes some inconsistent results reported by current studies, which leads to the objective of this study and is described in the last paragraph of the introduction part. As suggested by the reviewer, we deleted Section 3.1 in the revised manuscript. Key points from Section 3.1 are now included in the revised introduction (paragraph #2 and #3). Please refer to the revised manuscript.

- Model configuration: Despite the discussion of the importance of VOC emissions, the simulation period is 2018 whereas the MEIC emission inventory in China is 2017. I can understand the importance to understand the SOA modeling framework; however, emission status can be much more significant. In addition, S/IVOC emissions considered in this study targeted 2016. This also causes a difference in the simulation period, and also differ in the MEIC inventory. Is it ignoring the difference? I do not think so. Why the different year was applied and how can we understand the result caused by this difference?

  Response: We agree that it would be best to have consistent emission inventory for the modeling year. However, emission inventory data (MEIC and S/IVOC) for 2018 are not

publicly available, and the most recent year is 2017 and 2016, respectively. Due to the extensive efforts and time needed to update emission inventory, using earlier emission inventory to model more recent years is a common practice in the modeling community (e.g. Li et al. 2016; Han et al. 2016). In addition, while recognizing that using the same emission inventory and modeling years would be ideal, the uncertainties in the emission inventory itself are almost certainly larger than changes in emissions from 2017 to 2018. Therefore, we conclude that this difference does not substantially influence our major findings.

References:

Li, J., Zhang, M., Gao, Y., Chen, L., 2016. Model analysis of secondary organic aerosol over China with a regional air quality modeling system (RAMS-CMAQ). *Atmospheric and Oceanic Science Letters* 9(6), 443-450.

Han, Z., Xie, Z., Wang, G., Zhang, R., Tao, J., 2016. Modeling organic aerosols over east China using a volatility basis-set approach with aging mechanism in a regional air quality model. *Atmospheric Environment* 124, 186-198.

**Specific comments:**

- P1, L17-18: I feel AVOC and BVOC are sources whereas SVOC and IVOC are emission statuses, am I correct? Please clearly state this point. Do they simply summarize as "precursors"? In the case of BVOC, how can we suggest to policy makers? That is why the latter part of the abstract might introduce some confusion.

  Response: The definition of VOC, SVOC, and IVOC is based on the effective saturation concentration ($C^*$) under ideal or near ideal atmospheric conditions (298 K and 1 atm). VOC is the most volatile with a $C^*$ of $10^7$~$10^{11}$ µg/m$^3$; IVOC has intermediate volatility ($C^*$=$10^3$~$10^6$ µg/m$^3$), while SVOC has the lowest ($C^*$=$10^0$~$10^2$ µg/m$^3$) volatility among the three groups. VOC can be further divided into biogenic (BVOC) and anthropogenic (AVOC) based on the source. In contrast, SVOC and IVOC are always from anthropogenic sources; thus, they intrinsically mean anthropogenic SVOC and IVOC. All these groups are considered as SOA precursors, although the relative contributions differ by region and season. We re-wrote the emission inventory part (P6, L191-L200 in the revised manuscript) for better clarification.

  For policymakers, they would need to know the relative contribution of different OA components in order to make targeted emission reduction plans. In the case of BVOC, previous studies have shown that anthropogenic air pollutants, for example, primary organic aerosols (POA), SO$_2$, and NO$x$ have substantial impacts on SOA generated from

BVOC (BSOA; Carlton et al., 2010; 2018); thus some portion of BSOA are considered as "controllable". Besides, BVOC emissions could also be reduced directly by planting low-emitting species in built-up areas (Chang et al. 2012; Ren et al. 2017).

References:

Chang, J., Ren, Y., Shi, Y., Zhu, Y., Ge, Y., Hong, S., ... & Fu, C. (2012). An inventory of biogenic volatile organic compounds for a subtropical urban–rural complex. Atmospheric Environment, 56, 115-123.

Carlton, A. G., Pinder, R. W., Bhave, P. V., & Pouliot, G. A. (2010). To what extent can biogenic SOA be controlled?. *Environmental science & technology*, *44*(9), 3376-3380.

Carlton, A. G., Pye, H. O., Baker, K. R., & Hennigan, C. J. (2018). Additional benefits of federal air-quality rules: Model estimates of controllable biogenic secondary organic aerosol. Environmental science & technology, 52(16), 9254-9265.

Ren, Y., Ge, Y., Ma, D., Song, X., Shi, Y., Pan, K., ... & Chang, J. (2017). Enhancing plant diversity and mitigating BVOC emissions of urban green spaces through the introduction of ornamental tree species. Urban Forestry & Urban Greening, 27, 305-313.

- P1, L32-33: "ASOA (SOA produced from anthropogenic VOC)" should be defined in L28.

Response: In L28 "SOA generated from anthropogenic precursors" refers to SOA produced from AVOC, SVOC, and IVOC, not just ASOA, which is specifically defined as SOA produced from anthropogenic VOC (defined in L33-34).

- P1, L36: I cannot follow the wording "traditional VOC emissions" within this abstract.

Response: "Traditional VOC emissions" refers to VOC emissions that are usually accounted for in developing the emission inventory. This terminology is used to differentiate from S/IVOC emissions that are usually not accounted for in the emission inventory and has been used in many previous studies (examples are given below).

Hu et al. (2022): "*The total S/IVOC contribution generally contributes more than **traditional VOCs** under high NOx condition, however varied under low NOx condition.*"

Lu et al. (2018):"*For gasoline sources, we predict that IVOCs and SVOCs contribute as much SOA as **traditional VOC** precursors (mainly single-ring aromatics).*"

Deng et al. (2017):"*It demonstrated that **traditional VOC** precursors could not explain the amount of diesel SOA formation.*"

Chang et al. (2022): "***Traditionally, VOC** has been considered to be the only source of SOA…*"

References

Chang, X., Zhao, B., Zheng, H., Wang, S., Cai, S., Guo, F., ... & Donahue, N. M. (2022). Full-volatility emission framework corrects missing and underestimated secondary organic aerosol sources. *One Earth*, *5*(4), 403-412.

Deng, W., Hu, Q., Liu, T., Wang, X., Zhang, Y., Song, W., ... & George, C. (2017). Primary particulate emissions and secondary organic aerosol (SOA) formation from idling diesel vehicle exhaust in China. *Science of the Total Environment*, *593*, 462-469.

Hu, W., Zhou, H., Chen, W., Ye, Y., Pan, T., Wang, Y., ... & Wang, X. (2021). Oxidation flow reactor results in a Chinese megacity emphasize the important contribution of S/IVOCs to ambient SOA formation. *Environmental Science & Technology*, *56*(11), 6880-6893.

Lu, Q., Zhao, Y., & Robinson, A. L. (2018). Comprehensive organic emission profiles for gasoline, diesel, and gas-turbine engines including intermediate and semi-volatile organic compound emissions. *Atmospheric Chemistry and Physics*, *18*(23), 17637-17654.

- P1, L45: "OA" is not defined in the abstract.

  Response: "OA" refers to organic aerosol. Definition has been added to the abstract (P2, L44). In addition, to make it clearer, we have extended the title of this manuscript to "Modeling secondary organic aerosol contributions of VOC, IVOC and SVOC emissions and large uncertainties associated with organic aerosol aging".

- P2, L51-51: "more sophisticated measurement data and/or chamber experiments" for what? SOA aging? This sentence is needed to be rewritten.

  Response: Yes. Given the large differences of SOA due to different aging schemes, we suggest more sophisticated measurement data (e.g. observed data based on Aerosol Mass Spectrometer (AMS) that could resolved SOA from different sources) and chamber experiments (e.g. SOA yields) would be helpful to constrain the model parameterization of SOA aging. We rewrite the sentence as below:

  Revised manuscript (P2, L51-53):

  "More sophisticated measurement data (e.g. with resolved OA components) and/or chamber experiments (e.g. investigating how aging influences SOA yields) are needed to better characterize SOA aging and constrain model parameterizations."

- P2, L56-58: Where was indicated for these emission reductions, entire the world or China? It also needs the appropriate references to support them.

Response: The emission reductions are meant for China only. We added clarification as well as appropriate references in the revised manuscript (P2, L57-59):

"With the substantial emission reduction of sulfur dioxide (SO$_2$), nitrogen oxides (NO$x$), and primary particles in China over the past decade (Zheng et al. 2018; Yan et al., 2021), the organic portion of PM$_{2.5}$ becomes increasingly important".”

- P2, L58-61: However, this reference was older than 10 years. Does this support the increasing importance of OA?

Response: Thanks for pointing this out. We have updated with more recent references (P2, L64), Pye et al., 2021.

Pye, H.O.T., Ward-Caviness, C.K., Murphy, B.N., Appel, K.W., Seltzer, K.M., 2021. Secondary organic aerosol association with cardiorespiratory disease mortality in the United States. Nature Communications 12(1).

- P2, L63 or L65: No explanation for POA.

Response: POA refers to primary organic compound. Definition of POA was provided in P2, L62-63 of the revised manuscript.

- P2, L66: This 2nd paragraph in the introduction section is required to be reorganized. I would like to suggest just state emissions. The latter part of this paragraph contains other topics. Please also see the following specific comments.

Response: We thank the reviewer for providing specific comments. We have re-organized the whole introduction part. Please refer to our response to major comment #1.

- P3, L75: "missing SOA precursors" is ambiguous.

Response: "missing SOA precursors" refers to S/IVOC emissions since these compounds are usually not considered in traditional emission inventory. We revised the introduction part so this expression is not relevant.

- P3, L88: This sentence makes confusion. Does this CAMx model have more reliability to represent SOA modeling? As the authors introduced, the current models have two frameworks of two-product and VBS (from L93). Without the introduction to the modeling framework, I do not understand why this study was introduced abruptly.

Response: We are sorry for the confusion. This sentence was meant to support the statement that including IVOC emissions could increase simulated SOA concentration. We revised the sentence as below (P3, L100-102):

"For example, the inclusion of IVOC emissions was shown to increase the simulated SOA concentration by 5%-26% at an observation site in the Yangtze River Delta (YRD) region (Huang et al. 2021)."

- P3, L89-92: Again, what is the status of this WRF-Chem model? Why this study is introduced without a detailed introduction to the modeling framework? Moreover, this WRF-Chem model is not applied in this study. If the authors recognize this model, why this WRF-Chem model was not applied in this study?

Response: We are sorry for the confusion. Air quality models (e.g. CAMx, CMAQ, WRF-Chem, GEOS-Chem, RAQMS) contain many component algorithms to represent distinct physical (e.g. transport, diffusion, deposition) and chemical (e.g. photochemistry, aqueous chemistry, and heterogeneous reactions) processes. SOA frameworks/schemes, on the other hand, are just one among the many sub-modules implemented in air quality models to simulate SOA formation. Different air quality models could employ different SOA frameworks and the two-product and VBS schemes are two most commonly used SOA modules in current air quality models. For example, CAMx provides two options of simulating SOA, by choosing either the two-product or VBS scheme. CMAQ SOA module combines VBS framework for some SOA formation pathways and two-product for others. However, even several models may choose a VBS scheme to represent SOA chemistry, that does not mean that all VBS schemes are identical. Our study focuses on the SOA schemes rather than the host air quality model. By choosing CAMx and CMAQ, which are frequently applied in China due to their source apportionment features (e.g. Zhang et al. 2018; Li et al. 2016; Cao et al. 2022), we are able to compare two different VBS schemes. This approach met our objective of focusing on SOA schemes and produced clear findings on the importance of SOA aging assumptions. We have re-organized the introduction part to make the purpose clearer. The 2$^{nd}$ paragraph of the introduction describes commonly used SOA formation modules by different air quality models and their advantages and limitations. Please refer to the revised manuscript.

References:

Zhang, Y., Li, X., Nie, T., Qi, J., Chen, J., Wu, Q., 2018. Source apportionment of PM2.5 pollution in the central six districts of Beijing, China. Journal of Cleaner Production 174, 661-669.

Li, L., An, J.Y., Shi, Y.Y., Zhou, M., Yan, R.S., Huang, C., Wang, H.L., Lou, S.R., Wang, Q., Lu, Q., Wu, J., 2016. Source apportionment of surface ozone in the Yangtze River Delta, China in the summer of 2013. Atmospheric Environment 144, 194-207.

Cao, J.Y., Qiu, X.H., Peng, L., Gao, J., Wang, F.Y., Yan, X., 2022. Impacts of the differences in PM$_{2.5}$ air quality improvement on regional transport and health risk in Beijing-Tianjin-Hebei region during 2013-2017. Chemosphere 297.

- P3, L107-P4, L110: Is this DDM only applied in Houston, U.S.? If so, is it reliable to apply this method in China? I do not capture what is the intention to introduce this method.

  Response: The DDM method has been applied to many locations around the world (Du et al. 2022; Tang et al. 2015; Huang et al. 2020). However, it has only been used for the purpose of SOA source apportionment by the Path Integral Method (PIM) for Houston. The DDM and PIM are mathematical methods and therefore not tied to any one location. We introduced this method for completeness in describing the available source apportionment methods in addition to brute force and tagged species (PSAT).

  We re-organized the introduction part and feel these descriptions are not relevant to the current study so we removed this part in the revised manuscript.

  References:

  Du, X., Tang, W., Cheng, M., Zhang, Z., Li, Y., Li, Y., Meng, F., 2022. Modeling of spatial and temporal variations of ozone-NOX -VOC sensitivity based on photochemical indicators in China. Journal of Environmental Sciences 114, 454-464.

  Tang, W., Cohan, D.S., Pour-Biazar, A., Lamsal, L.N., White, A.T., Xiao, X., Zhou, W., Henderson, B.H., Lash, B.F., 2015. Influence of satellite-derived photolysis rates and NOx emissions on Texas ozone modeling. Atmospheric Chemistry and Physics 15(4), 1601-1619.

  Huang, R., Qin, M., Hu, Y., Russell, A.G., Odman, M.T., 2020. Apportioning prescribed fire impacts on PM2.5 among individual fires through dispersion modeling. Atmospheric Environment 223.

- P4, L113-115: I am further confused because this sentence cited the other model of GEOS-Chem. As the authors explained two modeling approaches, this and the following discussion for the difference of SOA concentration in China are unclear. What is the model used in Miao et al. (2021), An et al. (2022), Chang et al. (2022), and Wu et al. (2021)? This is related to the comment of Section 3.1, and I strongly suggest to re-organize this part for the review of previous studies. It is very hard to follow all of them.

  Response: As per suggested by the reviewer, we have deleted section 3.1 and re-organized the introduction section substantially. The model used in Miao et al. (2021), An et al. (2022), Chang et al. (2022), and Wu et al. (2021) refer to different air quality models,

which are all based on commonly used SOA modules (either the two-product or VBS, or combo of both). Please refer to our response above and the revised manuscript.

- P4, L127: Nevertheless of the introduction of WRF-Chem model and GEOS-Chem model, why the authors only applied two models of CAMx and CMAQ in this study? The reason is not clear.

Response: Both CAMx and CMAQ have been frequently used to address air pollution problems in China due to their ability of source apportionment for ozone and $PM_{2.5}$ (e.g. Zhang et al. 2018; Li et al. 2016; Cao et al. 2022). In terms of mechanisms, both models contain simulated mechanisms for SOA, including the volatility and aging mechanisms of POA. Secondly, as commonly used air quality models, there is no other article that compares the simulation of SOA between the two. We have revised the sentence as below (P4, L126-131):

"CAMx and CMAQ are two frequently used models to address various air pollution questions in China, partially due to their abilities to track source contribution (e.g. Zhang et al. 2018; Li et al. 2016; Cao et al. 2022). Both models incorporated the VBS scheme to simulate SOA but their simulations results have not been compared with each other. Therefore, this study attempts to resolve some of the existing inconsistencies of simulated SOA by a detailed comparison between two commonly used air quality models – CAMx and CMAQ."

References:

Zhang, Y., Li, X., Nie, T., Qi, J., Chen, J., Wu, Q., 2018. Source apportionment of PM2.5 pollution in the central six districts of Beijing, China. Journal of Cleaner Production 174, 661-669.

Li, L., An, J.Y., Shi, Y.Y., Zhou, M., Yan, R.S., Huang, C., Wang, H.L., Lou, S.R., Wang, Q., Lu, Q., Wu, J., 2016. Source apportionment of surface ozone in the Yangtze River Delta, China in the summer of 2013. Atmospheric Environment 144, 194-207.

Cao, J.Y., Qiu, X.H., Peng, L., Gao, J., Wang, F.Y., Yan, X., 2022. Impacts of the differences in $PM_{2.5}$ air quality improvement on regional transport and health risk in Beijing-Tianjin-Hebei region during 2013-2017. Chemosphere 297.

- P4, L127-129: Repeatedly, we can easily find the application of WRF-Chem and GEOS-Chem model in China (also as shown in Table 2), so I do not figure out the approach taken in this study. Why only CAMx and CMAQ was applied? Is it enough to answer the authors' motivation to clarify the SOA modeling?

Response: We are sorry for the confusion. This comment is similar to previous ones. Please refer to our response above. We have also revised our introduction to more clearly explain our study objectives and design. Please refer to the revised manuscript.

- P7, L211-212: In the case of the inclusion of S/IVOC emissions, is it excluded a possibility of double-count in conventional VOC emissions?

Response: There is a potential for double counting organic emissions (e.g., between SVOC and PM emissions). This must be considered in the development of the emission inventory (Wu et al. 2021). In our study, we adopted the conventional VOC emissions from MEIC, and S/IVOCs emissions developed by Wu et al. (2021). We believe that the potential double-counting has been well addressed in previous studies. We re-arranged this part in the revised manuscript to avoid confusion (P6, L191-L200):

"Conventional anthropogenic VOC emissions were provided in the MEIC inventory while biogenic VOC emissions were calculated using a recent offline version of the Model of Emissions of Gases and Aerosols from Nature (version 3.2, https://bai.ess.uci.edu/megan). Specifically, MEIC provides VOC emissions speciated for Carbon Bond 2005 (CB05) and SAPRC 2007 (SAPRC07), which were combined to generate speciated VOC emissions for the CB6 mechanism (see Supplemental text 2). To evaluate the contributions of SVOC and IVOC emissions to SOA formation, we included a gridded ($0.25° \times 0.25°$) monthly S/IVOC emission inventory from industry, residential, transportation, power plants, and shipping for the year 2016, which was developed by applying source-specific emission ratio of S/IVOC to POA (Wu et al., 2021a)"

References

Wu, L., Ling, Z., Liu, H., Shao, M., Lu, S., Wu, L., Wang, X., 2021. A gridded emission inventory of semi-volatile and intermediate volatility organic compounds in China. Science of the Total Environment 761.

- P8, Section 3.1 and Table 2: This is not the results and discussions in this study. I would like to suggest shortening this section to highlight the points related to this study, and then moving this section to the introduction or supplemental materials. Moreover, this review contains both modeling approaches and uncertainty in emissions. It is much more readable to possibly divide based on these two viewpoints.

Response: We thank the reviewer for the constructive suggestions. We deleted Section 3.1 in the revised manuscript and moved Table 2 to the supplemental materials. Key points related to this study are included in the revised introduction ($2^{nd}$ and $3^{rd}$ paragraph of the introduction). We substantially re-organized the introduction part: the $2^{nd}$ paragraph of the

revised introduction now discusses the SOA modeling approaches and the 3rd paragraph of the introduction discusses the uncertainty in S/IVOC emissions. Please refer to the revised manuscript.

- P11, Section 3.2: I do not agree with the organization of this section. The authors applied the step-by-step scenarios as _base, _IVOC, and _S/IVOC; hence, the model evaluation should also be step-by-step. Why only _S/IVOC results are shown? We can expect the improvement of modeling performance in _S/IVOC rather than _base and _IVOC; however, if there is a degradation in modeling performances, what brings this study approach?

  Response: We have added step-by-step evaluation results in the revised supporting information (Table S10). Since $PM_{2.5}$ concentration increased with the addition of S/IVOC emissions, the model performance improved over regions that were underestimated in base scenario but degraded over regions that were overestimated in the base scenario. For example, $PM_{2.5}$ over NCP in the base CMAQ simulation was largely underestimated with a NMB of -28.3% for July and -23% for November. Adding S/IVOC emissions reduced the NMB to -16.0% and -11.4% for July and November, respectively. In contrast, $PM_{2.5}$ over NCP in the base CAMx simulation was already overestimated for July (NMB = 9.1%), adding S/IVOC emissions resulted an even higher NMB of 27.5%. However, considering that there are many uncertainties in both emission inventories and model algorithms for OA, we do not necessarily expect that every change will improve model performance for this one modeling episode. Rather, we are interested in understanding how emission inventories and model algorithms influence model performance in the hope that these results, together with others, will guide modeling system improvements.

- P11, Figure 2: What indicates a high concentration in the bottom-left corner in (c) and (d)? Is it long-range transport? But southwest China posed a lower level of $PM_{2.5}$. Is it a high concentration in other countries? Why the map was not shown? This figure itself posed confusion.

  Response: This is due to high emissions from the surrounding countries, e.g. India. We have added country boundaries to the plots. Please refer to revised Figure 2.

- P11, L311-P12, L315: In this context, CAMx and CMAQ mean CAMx_S/IVOC and CMAQ_S/IVOC listed in Table 1, right? It is better to explicitly identify the model name (like L294).

  Response: Thanks for pointing this out. In this context, CAMx and CMAQ do not necessarily mean 'CAMx_S/IVOC' and 'CMAQ_S/IVOC' scenarios. The different OA

components are calculated either as direct model outputs (for example, POA) or as differences between different scenarios. For example, IVOC-SOA is calculated as the differences between 'model_IVOC' and 'model_base' and SVOC-SOA is calculated as the differences between 'model_S/IVOC' and 'model_IVOC'. So here CAMx and CMAQ mean the model itself not just the 'CAMx_S/IVOC' and 'CMAQ_S/IVOC' scenarios.

- P12, L324 and hereafter: The wording such as "CMAQ PRD" and "CAMx Sichuan Basin" are not understandable. Please rewrite these wording.

  Response: Thanks for pointing this out. We have rephrased these expressions in the revised manuscript (P9, L284-285, L288-289):

  "In July, POA ranged from 0.7 μg/m$^3$ in PRD (simulated by CMAQ) to 4.1 μg/m$^3$ in Sichuan Basin (simulated by CAMx)…"

  "In November, domain-averaged POA ranged from 2.7 μg/m$^3$ in PRD (simulated by CMAQ) to 12.9 μg/m$^3$ in Northeast (simulated by CAMx)…"

- P12, L334-335: We can expect this BVOC status, but cannot fully understand it as input data in this study. I think the need to explicitly show the emission itself used in this study.

  Response: Thanks for pointing this out. We added a spatial plot showing the BVOC emissions in the supplemental information (Figure S5) to facilitate understanding.

- P12, L331: In this subsection, the relative percentage was well discussed; however, we do not follow these values from Fig. 3. I would like to suggest preparing the supplemental figure for the relative percentage (as spatial distribution).

  Response: Thanks for pointing this out. Figure 4 and 5 gives relative percentage for each sub-region, which the reader should be referring to. We added clarifications (P9, L272-274) to help readers understand the results shown in Figure 4 and 5. Spatial distributions of relative percentage are also provided in the supplemental information (Figure S4) for better illustration.

- P13, L361-365: This discussion might be understandable; however, it could be confirmed by applying the older version of CMAQ. I can partly agree that the scope of this study is on anthropogenic emissions; however, the portion of anthropogenic and biogenic are important points as presented in this study. Does this also imply that the model framework and configuration in CMAQ can cause differences in anthropogenic/biogenic sources?

  Response: Air quality models evolve continuously to incorporate new findings, for example, new formation mechanisms, new precursors, and updated model parameters. However, running older model versions would add complexity to our study. In addition,

since model updates are not just confined to the SOA scheme, the interpretation of results between different model versions may not be straightforward. Our view is that current model versions are most relevant and our aim is to help modelers to understand the similarities and differences between current versions of the two models.

- P14, Figure 3: Same point for Figure 2. What is a high concentration outside China?

Response: This is due to emissions outside China. We have added country boundaries in the revised Figure 3.

- P15, Figs. 4 and 5: The light blue color seems to be out of alignment. Is it a corrupted figure? Please confirm. The wording "IVOCs" and "SVOCs" did not correspond to the main text. It is better to unify.

Response: This is not a corrupted figure. We intentionally made the light blue color out of alignment to differentiate between POA (light blue color) and SOA (i.e. ASOA, BSOA, IVOC-SOA, SVOC-SOA). The total of different SOA components sums up to 100% while the total of POA and SOA sums up to 100%. This is also noted in the figure title: *"Note that the relative contribution of ASOA, BSOA, IVOC-SOA, and SVOC-SOA sum to 100%".* We have corrected the labels ("IVOCs" to "IVOC" and "SVOCs" to "SVOC") to be consistent throughout the main text.

- P16, L390: Again, do CAMx and CMAQ means CAMx_S/IVOC and CMAQ_S/IVOC listed in Table 1? It should be explicitly mentioned.

Response: No. In this situation, CAMx and CMAQ means the model itself, instead of the simulation scenarios listed in Table 1. We are sorry for this confusion and have added "CAMx and CMAQ models" to be explicitly referring to the model itself (P14, L352).

- P16, L397: We do not follow the detail of this RAQMS model from this manuscript.

Response: We revised the sentence to only showing the key point we want to make, instead of introducing unnecessary information (now P14, L357-359):

"The relatively high contribution of ASOA in CAMx (35.6%~59.1% of total SOA) is consistent with the study by Li et al. (2022), who reported 48.3% of SOA was formed from AVOC emissions."

- P17, L430: What are the explicit definitions for "high-" and "low-" NOx conditions? Are there some values to divide them?

Response: By convention, "high-NOx" refers to conditions where $RO_2$ radicals react primarily with NO whereas under low-NOx conditions they react primarily with $HO_2$

radical (Seinfeld and Pandis, 1998). We have added this clarification in the revised manuscript (P15, L385-388).

References:

Seinfeld, J. H., & Pandis, S. N. (1998). *Atmospheric chemistry and physics: from air pollution to climate change*. John Wiley & Sons.

- P17, L432: I do not fully understand the wording "outside the models". Is it a standalone SOA model from CAMx and CMAQ? How can these be outside?

  Response: Thanks for this comment. "outside the models" means calculations were performed with a standalone SOA model that solely computes SOA yields based on Pankow's partitioning theory, the total OA concentration and temperature. Under this circumstance, we are not accounting for physical (i.e. vertical diffusion, horizontal transport, dry deposition) and chemical (i.e. photolysis, radical oxidation) processes that are accounted for when SOA are calculated "within the model". We added this clarification in the revised manuscript (P14, L378-381):

  "We investigated how the different treatments of SOA aging in the CMAQ and CAMx VBS schemes influence simulated SOA concentrations by performing simple conceptual calculations outside the models. By doing this, the SOA yields are computed solely based on Pankow's partitioning theory, the total OA concentration and temperature. Other physical processes (e.g. removal by deposition, atmospheric dispersion) or chemical reactions (i.e. complex radical chemistry) are not included."

- P17, L443: "CMAQ_no-aging" is unlabeled "CMAQ" in Fig. 7? It is better to be unified.

  Response: Thanks for pointing this out. We unified the labels in Figure 6. Now the four cases are "CMAQ_no-aging", "CMAQ_aging" (previously "CMAQ_OLIG"), "CAMx_no-aging", and "CAMx_aging".

- P17, L448: To be consistent with Fig. 6, this should be "CMAQ_OLIG". Which is correct?

  Response: Thanks for pointing this out. We renamed "CMAQ_OLIG" to be "CMAQ_aging".

- P17, L462: There is no Table 6. Does this mean "Figure 6"?

  Response: Thanks for pointing this out. This is a typo. It should be "Table S13".

- P18, L460: According to this sentence, N in Eq. (1) is 5?

  Response: Yes. N is 5. We added the clarification in revised manuscript (P15, L385).

**Technical corrections:**

- P3, L87: No need to use the parenthesis because this reference was used as a subject.

  Response: Corrected in revised manuscript.

- P4, L117: The typo of "Miao's (2021)"?

  Response: This should be "Miao's (2021)". We have corrected in the revised manuscript (P4, L113).

- P9, L272: I do not find the reference of Miao et al. (2017), Typo in a year?

  Response: Thanks for pointing this out. It's a typo. This should be "Miao et al. (2021)" and has been corrected in the revised supplemental information.

**Reviewer #2**

This manuscript provides insight into model predictions of SOA over China using two common regional models: CAMx and CMAQ. The ability to find robust messages, such as the important role of IVOCs, across diverse model representations is useful. However, the current manuscript insufficiently describes the base models and should go farther in providing insight into what is well represented vs not.

Response: We thank the reviewer for affirming the value of this manuscript and providing helpful comments. We have revised the manuscript substantially to incorporate the reviewer's comments. Our major revisions include: (1) added descriptions of CMAQ SOA modules with appropriate citations (P5, L175-179); (2) added more discussions on the comparison between CMAQ and CAMx (P5, L179-P6, L189); (3) moved Section 3.1 to supplemental information with main points included in the introduction part. Please see our point-to-point response below.

**Major comments:**

1. Provide a better description of CMAQ SOA model with appropriate citations.

    o CMAQ AERO7 uses a VBS treatment for some systems, but not all. Specifically, many aqueous pathways and the "pcSOA" approach are not VBS style. Relabel the CMAQ treatment from "CMAQ VBS" to "CMAQ AERO7" throughout the manuscript to be more complete.

    Response: Thanks for pointing this out. We agree with the reviewer that "CMAQ AERO7" is more appropriate thus we have modified "CMAQ VBS" to "VBS in CMAQ AERO7" in the revised manuscript.

    o Several SOA pathways in CMAQ are not mentioned and the current description does not adequately cite CMAQ developments (none of the biogenic SOA articles are cited, for example). Provide a more complete description of SOA in CMAQ vs CAMx and add references as appropriate (See https://www.epa.gov/cmaq/how-cite-cmaq). Consider:

    ▪ CMAQ IEPOX SOA approach (not cited)

    ▪ CMAQ oligomer approach (not cited)

    ▪ CMAQ organic nitrate SOA (not mentioned nor cited)

    ▪ CMAQ monoterpene photooxidation SOA (mentioned but not cited)

    ▪ CMAQ semivolatile POA approach (could be better cited)

- CMAQ glyoxal/methylglyoxal SOA approach (neither mentioned nor cited)

Response: Thanks for pointing this out. We have added a more complete description of SOA formation in CMAQ with appropriate references. Please see revised manuscript (P5, L175-P6, L189).

o One of the conclusions is: "CMAQ tends to estimate higher BSOA concentration, while CAMx generates more ASOA." You could mention that CMAQ has more pathways to SOA from biogenic precursors, including aqueous pathways that are not present in CAMx.

Response: Thanks for the suggestion. We have added this in the revised manuscript (P10, L323-L328).

o Emissions and model choices (e.g., MEGAN) on lines 151-160 could also use literature citations.

Response: Modified in the revised manuscript (P5, L158).

o Is biogenic SOA in CAMx subject to VBS aging?

Response: No. Aging of biogenic SOA is disabled CAMx based on previous modeling studies that found aging biogenic SOA led to a significant over-prediction of OA in rural areas (Lane et al., 2008; Murphy and Pandis, 2009). We added this clarification in P6, L183-L184.

References

Lane, T.E., Donahue, N.M., Pandis, S.N., 2008. Simulating secondary organic aerosol formation using the volatility basis-set approach in a chemical transport model. Atmospheric Environment 42(32), 7439-7451.

Murphy, B.N., Pandis, S.N., 2009. Simulating the Formation of Semivolatile Primary and Secondary Organic Aerosol in a Regional Chemical Transport Model. Environmental Science & Technology 43(13), 4722-4728.

2. Can the authors go further in determining whether parametrizations are realistic and what improvements might be needed?

o Figure 6: Is there experimental data to support one parameterization over another? Have you compared the CAMx aging scheme to more recent 2-D VBS parameterizations such as the work of Zhao et al. (2016)? Given 5.75 hours of aging could be captured in a chamber experiment is there data to confirm a 30x increase in yield? Is the aging scheme in CAMx plausible?

Response:

(1) Using chamber experimental data to validate the SOA aging is a good idea but the results are difficult to interpret. The increased SOA mass simulated here is purely due to the aging of SOA itself. Whereas in chamber experiments, increased SOA mass is due to both aging of SOA and freshly formed SOA from precursors. We hope the strong sensitivity to aging scheme found here will guide experimentalists toward conducting studies that are most useful for constraining current modeling approaches

(2) We compared the CAMx results in our study with Zhao et al (2016), which is illustrated by Figure R1 and R2. Note that the year (2010) and month (August and November) simulated in Zhao et al. (2016) is not exactly same as this study (2018 July and November). In general, the spatial and seasonal patterns between Zhao's results and our results are similar. In terms of magnitude, Zhao simulated lower SOA in summer and much higher SOA in November. However, since SOA yields are sensitive to the POA concentration, with higher POA concentration increasing the SOA yield, these SOA differences can be attributable to different POA emissions, different SOA schemes, or both.

Analyzing OA by regions (note that the regions defined in Zhao's study are not exactly same as this study), CAMx simulated higher ASOA than Zhao's 2D VBS, in July, particularly over NCP and YRD, which is consistent with more aging effects with the CAMx VBS scheme. For BSOA, simulated BSOA in Zhao's study is significantly lower than our results (simulated by both CAMx and CMAQ). One possible reason for this is an older version of MEGAN (v2.04) was used in Zhao's study while MEGANv3 was used in our study. A direct comparison of the BVOC emissions would be needed to confirm this. For POA, Zhao simulated much higher values than this study, probably due to much higher emissions back in 2010 compared to the 2017 emissions used in this study. For IVOC-SOA, our results are slightly lower or comparable to that of Zhao. In November, results in this study look more similar to Zhao's except POA and IVOC-SOA is much higher in Zhao's study, given that emissions were higher in 2010 than 2017.

[Figure]

Figure R1 Spatial distribution of monthly averaged SOA concentration (µg/m³) in Zhao et al. (2016) and this study

[Figure]

Figure R2 Comparison of simulated OA components by regions and by month where "CMAQ H-VBS" shows results from Zhao et al. (2016).

(3) The aging effect calculated by the off-line conceptual model is just an illustration of the aging differences between CAMx and CMAQ, without considering other processes, such as vertical and horizontal transport. Therefore, a factor of 30 is just a theoretical number that is not directly comparable to chamber experiments because wall-losses are an important factor in chamber experiments that isn't relevant to our conceptual experiments. To demonstrate the effect of SOA aging in real 3D application, we conducted another set of CAMx base simulation (i.e. without S/IVOC emissions) with no ASOA aging by setting the rate constant of OH with SOA to zero. The spatial distribution of ASOA with and without aging is displayed in Figure R3. We further calculated the effect of aging by region and compared to Zhao's results in Table R1. According to Zhao's results,

the aging of SOA from AVOC in their 2D-VBS increased the SOA mass by a factor of 2~3.7 times compared to the two-product scheme in CMAQv5.0.1 over different regions. In contrast, the aging effect in CAMx 1.5D VBS increased ASOA mass by factors of 6~14.3 in July and 6.3~9.3 in November, which is on average 2~3 times stronger. These discussions have been added to the revised manuscript (P14, L375-377; P16, L423-433).

Table R1 Comparison of aging effect in Zhao et al. (2016) and this study

| Month | AVOC-SOA increase factor due to aging | NCP[b] | YRD | PRD | Sichuan Basin |
|---|---|---|---|---|---|
| July/August | This study | 14.3 | 9.7 | 11.0 | 6.0 |
| | Zhao et al. (2016)[a] | 3.3 | 3.3 | 3.3 | 2.0 |
| November | This study | 6.3 | 8.2 | 9.3 | 7.0 |
| | Zhao et al. (2016) | 2.0 | 3.7 | 3.0 | 2.3 |

[a] In Zhao's study, increase factor is calculated as the ratio of "H-VBS" scenario to "CMAQ" scenario.

[b] Note that the definition of regions are not exactly same in Zhao's study and this study

[Figure]

Figure R3 Spatial distribution of ASOA for aging and no-aging in CAMx base scenarios

References

Zhao, B., Wang, S., Donahue, N.M., Jathar, S.H., Huang, X., Wu, W., Hao, J., Robinson, A.L., 2016. Quantifying the effect of organic aerosol aging and

intermediate-volatility emissions on regional-scale aerosol pollution in China. Scientific Reports 6.

o The text on line 401 through 417 (differences in the CMAQ vs CAMx benzene system) should be refocused. The experimental data (Ng et al., 2007) demonstrates why there is only mass in the lowest C* bin for CMAQ—the SOA was observed to be nonvolatile in terms of yield behavior. Providing the total yield of C* 1000 ug/m$^3$ and lower species in CAMx vs CMAQ confuses the story which is better captured on the following page. Table S9 indicates CMAQ would predict an SOA yield from benzene of 0.146 for all atmospherically relevant conditions. CAMx would predict an SOA yield of about 10% (loading of 10ug/m$^3$ assumed) so the CAMx SOA yield for benzene before any multigenerational aging is lower than CMAQ (later shown in Fig 6). This suggests that it isn't the initial benzene SOA yields (whether wall loss corrected are not) that are driving differences.

Response: We agree with the reviewer. We removed these texts in the revised manuscript.

o Reword sentences to bring clarity and specificity. For example: "Our results indicate that large uncertainties still exist in the simulation of SOA in current air quality models due to the aging schemes as well as uncertainties of the emission inventory" can be reworded to: "Our results indicate aging schemes are the major driver in CMAQ vs CAMx treatments of ASOA and their resulting predicted mass." (The role of emission inventories wasn't specifically addressed and could be removed.)

Response: Thanks for the suggestion. We revised this sentence accordingly (P2, L49-L50):

"In addition, aging schemes are the major driver in CMAQ vs. CAMx treatments of ASOA and their resulting predicted mass."

We have also made extensive edits throughout the manuscript by incorporating the reviewer's comments. In addition, we have re-organized the introduction part with the following order: the 1$^{st}$ paragraph presents the importance of SOA as before. The 2$^{nd}$ paragraph now describes commonly used SOA formation modules by different air quality models and their advantages and limitations. The 3$^{rd}$ paragraph discusses the importance of S/IVOC emissions as SOA precursors. Section 3.1 was deleted and Table 2 was moved to the supplemental information.

o Can the regional model bias be used to help inform which representations are plausible? For example, what is the spatial pattern of bias? Since BSOA and ASOA have some spatial separation, how does performance very by model species? Observations could be added to Fig 4.

Response: Thanks for the comment. However, since we don't have observed OA components, we cannot evaluate model performance by individual OA component. The table below shows the ranges of SOC MB and NMB over NCP, YRD and PRD regions, where OC/EC observations are available. A general trend is that in for July, models tend to overestimate SOC over northern regions (i.e. NCP) and underestimated over southern regions (i.e. PRD) whereas the opposite is true for November. However, without having observed OA components or precursor measurements, interpreting these trends would be speculative because, for example, regional variation in biomass composition and therefore BVOC emissions could be confounded with biases within the SOA schemes.

We cannot add observation to Figure 4 because we only have OC observations at limited stations whereas Figure 4 shows the regional averages.

Table R2 SOC MB and NMB by regions

| | CAMx_S/IVOC | | | CMAQ_S/IVOC | | |
|---|---|---|---|---|---|---|
| | NCP | YRD | PRD | NCP | YRD | PRD |
| July MB ($\mu g/m^3$) | 3.1~5.5 | 0.6~3.0 | -1.0~0.3 | 0.1~2.3 | -0.3~1.3 | -1.4~-0.3 |
| July NMB (%) | 72~156 | 10~108 | -30~15 | 1~66 | -6~34 | -42~-12 |
| November MB ($\mu g/m^3$) | -2.0~0.1 | 0.6~3.5 | / | -1.3~0.6 | 0.5~3.3 | / |
| November NMB (%) | -31~1 | 18~83 | / | -19~14 | 13~79 | / |

3. The authors map IVOC emissions in CMAQ to the pcSOA precursor (pcVOC). From Murphy et al. (2017): "We further introduce a new surrogate species, potential SOA from combustion emissions (pcSOA) to account for missing mass from IVOC oxidation, multigenerational aging of (anthropogenic) secondary organic vapors (from IVOC and VOC precursors), biases in SOA yields from vapor wall losses, and enhanced organic partitioning to the condensed aqueous phase. In addition to these sources, pcSOA could account for mass from oxidation of as-yet unidentified sources of SOA precursors." Are IVOCs a good fit for the pcSOA precursor? How much does the emission magnitude of IVOCs differ from what Murphy et al. proposed as the emission rate (which was not IVOC specific)? How does the yield of SOA from pcSOA compare to that expected for IVOCs?

Response: Thanks for pointing this out. IVOC emissions used in this study include residential, transportation, power plants, shipping, and industry (Wu et al. 2021), which represent the combustion IVOC emissions as opposed to the non-combustion IVOC (e.g. volatile chemical products). Table R3 contrasts the IVOC emissions used in this study and the pcVOC emissions calculated based on POA scaling factor provided with default CMAQ (a factor of 6.579). While other sectors are of similar emission magnitudes, the biggest difference lies in the residential sector, where pcVOC emissions are 8 times higher in July and more than 20 times higher in November. As a result, the total pcVOC emissions are higher than IVOC emissions by a factor of ~2 in July and ~4 in November. IVOC emissions from residential combustion depend on the fuel types and combustion conditions and could vary in a wide range (Cai et al. 2019; Qian et al. 2021). Qian et al. (2021) reported a value of 175.89 Gg/year for the annual IVOC emissions from residential solid fuel combustion in 2014 based on field measurement in rural China. Zheng et al. (2023) reported a decreasing trend of residential IVOC emissions (including residential biomass and residential fossil fuel) from 2.29 Tg/year in 2005 to 1.15 Tg/year in 2019. These values are closer to the IVOC emissions used in this study whereas the pcVOC emissions are too large.

Table R3 Emissions of pcVOC and IVOC by sector and month (percentages in the brackets represent relative contribution)

| Month | Sector | pcVOC (Tg) | IVOC (Tg) |
|---|---|---|---|
| | Industry | 0.2 (19.1%) | 0.3 (70.6%) |
| | Power | $3.6\times10^{-6}$ (0.0%) | $4.5\times10^{-6}$ (0.0%) |
| Jul. | Residential | 0.8 (74.3%) | 0.1 (10.5%) |
| | Transport | 0.1 (6.6%) | 0.1 (19.0%) |
| | *Total* | *1.1* | *0.5* |
| | Industry | 0.2 (9.3%) | 0.4 (67.8%) |
| | Power | $3.5\times10^{-6}$ (0.0%) | $4.0\times10^{-6}$ (0.0%) |
| Nov. | Residential | 2.3 (87.8%) | 0.1 (16.5%) |
| | Transport | 0.1 (2.8%) | 0.1 (16.1%) |
| | *Total* | *2.6* | *0.6* |

We further conducted two CMAQ simulations using the pcVOC emissions instead of the IVOC emissions. Figure R4 shows the spatial distribution of SOA contribution due to IVOC (i.e. IVOC-SOA) and pcVOC (i.e. pcSOA) and Table R4 shows the corresponding sub-domain averages. As expected, psSOA is much higher than IVOC-SOA because pcVOC emissions are much higher than IVOC emissions. In July, pcSOA is higher by 1.67 (YRD) ~ 2.5 times (Sichuan Basin) than IVOC-SOA. In November, the ratio is even higher (2.4 in PRD to 5 in Northeast). We added this comparison in the supplemental information (supplemental text 3).

[Figure]

Figure R4 Spatial distributions of pcSOA (left column) and IVOC-SOA (right column) in July (upper row) and November (bottom row)

Table R4 Domain averaged SOA generated from pcVOC (i.e. pcSOA) and IVOC (i.e. IVOC-SOA) emissions (unit: $\mu g/m^3$)

| Month | Region | pcSOA | IVOC-SOA |
|-------|--------|-------|----------|
| July | Northeast | 2.8 | 1.8 |
| | NCP | 7.0 | 4.0 |
| | YRD | 3.5 | 2.1 |
| | PRD | 1.7 | 0.8 |
| | Central China | 4.0 | 1.8 |
| | Sichuan Basin | 5.3 | 2.1 |
| November | Northeast | 7.0 | 1.4 |
| | NCP | 11.3 | 3.9 |
| | YRD | 12.4 | 5.0 |
| | PRD | 8.0 | 3.4 |
| | Central China | 13.4 | 5.1 |
| | Sichuan Basin | 10.9 | 3.0 |

References

Cai, S., Zhu, L., Wang, S., Wisthaler, A., Li, Q., Jiang, J., & Hao, J. (2019). Time-resolved intermediate-volatility and semivolatile organic compound emissions from household coal combustion in Northern China. Environmental Science & Technology, 53(15), 9269-9278.

Murphy, B.N., Woody, M.C., Jimenez, J.L., Carlton, A.M.G., Hayes, P.L., Liu, S., Ng, N.L., Russell, L.M., Setyan, A., Xu, L., Young, J., Zaveri, R.A., Zhang, Q., Pye, H.O.T., 2017. Semivolatile POA and parameterized total combustion SOA in CMAQv5.2: impacts on source strength and partitioning. Atmospheric Chemistry and Physics 17(18), 11107-11133.

Qian, Z., Chen, Y., Liu, Z., Han, Y., Zhang, Y., Feng, Y., ... & Tao, S. (2021). Intermediate volatile organic compound emissions from residential solid fuel combustion based on field measurements in rural China. Environmental Science & Technology, 55(9), 5689-5700.

Zheng, H., Chang, X., Wang, S., Li, S., Yin, D., Zhao, B., ... & Xing, J. (2023). Trends of Full-Volatility Organic Emissions in China from 2005 to 2019 and Their Organic Aerosol Formation Potentials. Environmental Science & Technology Letters, 10(2), 137-144.

**Minor comments:**

1. Clarify your definition of biogenic (BSOA and BVOC). Are BVOCs strictly from vegetation or defined as specific VOCs such as isoprene and monoterpenes? If BVOCs are defined based on isoprene or monoterpene identity, please highlight that anthropogenic monoterpene emissions can be substantial (Coggon et al., 2021) and anthropogenic NOx modulates monoterpene SOA (Pye et al., 2015) and thus biogenic does not mean the SOA is entirely biogenic.

   Response: Thanks for pointing this out. BVOC in this study refers to the VOC emissions emitted from vegetation, not defined as specific VOCs such as isoprene or monoterpenes. BVOC is calculated by MEGAN and includes isoprene, monoterpenes, sesquiterpenes and etc. Similarly, BSOA refers to SOA formed from VOCs emitted by vegetation (mainly isoprene, monoterpenes, and sesquiterpenes). We added this clarification in revised manuscript (P5, L168-170).

2. Line 101: What is meant by brute-force SOA estimation? Is that a zero out?

   Response: Yes, it is a zero out method. However, per suggestion from the other reviewer, we re-organized the introduction part substantially by incorporating key points from Section 3.1 (Section 3.1 was deleted). In the revised introduction, we described commonly used SOA formation modules by different air quality models and their advantages and limitations and then discussed the importance of S/IVOC emissions as SOA precursors This comment is not relevant for the revised manuscript.

3. Line 125: I recommend removing "No clear conclusions can be drawn." Often, the different results reflect different model parameterizations. The reason they are giving different answers can be (at least partially) identified.

Response: Thanks for pointing this out. We deleted this sentence in the revised manuscript.

4. Section 2.2: Include a brief overview of how emissions from previous were developed (were they scaled to POA)?

Response: Thanks for the suggestion. The S/IVOC emissions were developed based on source-specific ratio of S/IVOC to POA (Wu et al. 2021). We added descriptions in the revised manuscript (P6, L198-L199):

"… we included a gridded (0.25°×0.25°) monthly S/IVOC emission inventory from industry, residential, transportation, power plants, and shipping for the year 2016, which was developed by applying source-specific emission ratio of S/IVOC to POA (Wu et al., 2021a)."

References:

Wu, L., Ling, Z., Liu, H., Shao, M., Lu, S., Wu, L., Wang, X., 2021a. A gridded emission inventory of semi-volatile and intermediate volatility organic compounds in China. Science of the Total Environment 761.

5. Line 193: Are these percents of total S/IVOC or total VOC?

Response: These are percent of total S/IVOC. We added clarification in the revised manuscript (P6, L201-L203):

"On an annual scale, industrial and residential sources accounted for 12.6% and 64.6% of the total SVOC emissions and 63.1% and 15.5% of the total IVOC emissions."

6. The 1.6 OM/OC ratio attributed to Feng et al. is actually from Turpin and Lim 2001. The value seems a bit low considering primary wood burning emissions often have OM/OC ratios of 1.7. Consider updating the OM/OC from 1.6 to a more recent value. Alternatively, model output can be converted to OC as the model often has a specific molecular weight and other properties assigned to the species. CMAQ specifies OM/OC ratios in the Species Definition files supplied with the model (https://github.com/USEPA/CMAQ/blob/5.3.2/CCTM/src/MECHS/cb6r3_ae7_aq/SpecDef_cb6r3_ae7_aq.txt).

Response: Thank you for the suggestion. We updated the validation of SOA to SOC accordingly. Please refer to revised manuscript.

7. Section 3.1: Experimental data to feed parameterizations has increased in concentration range over time which is what allows a greater range of volatility to be fit in the VBS vs older data sets. Similarly, older data tended to be from experiments from very high loading which made extrapolation to ambient atmospheres more difficult and likely drove errors. Consider adding this context.

Response: We understand this comment as providing context for why VBS schemes tend to be derived from experiments that are more atmospherically relevant than were used to develop older "two-product" schemes. However, since both models have VBS schemes this context doesn't seem to fit into this manuscript.

8. Line 364: Cite peer-reviewed original references rather than model release notes.

Response: Revised in the manuscript (P10, L326).

9. Line 417: Remove personal communication citation. The CMAQ benzene yields can be traced back to experimental data which indicates if vapor wall loss was performed.

Response: This part has been removed in the revised manuscript so this comment is not relevant.

10. Figure 6: Add aging time of 5.75 hours to caption.

Response: Modified in the revised manuscript.

11. At least one critical reference is missing from Table 1 (Zhao et al., 2016).

Response: Per suggestion from the other reviewer, we moved this table to supplemental information and added all cited references.

12. Reword the citation on line 64-65—the reason for the association with SOA and mortality has not been determined.

Response: Thanks for pointing this out. We modified the words in the revised manuscript (P2, L66-L67):

"A recent study by (Pye et al., 2021) reported that SOA is associated with higher rate of cardiorespiratory mortality than $PM_{2.5}$ on a per mass basis."

13. Table 1 could be moved to the SI. Also consider relabeling as some figure labels use the Table 1 labels (Fig 2) and others do not (Fig 3).

Response: Thanks for pointing this out. We moved Table 1 to SI. Figure 3 did not use the Table 1 labels because "CAMx" and "CMAQ" in Figure 3 do not present the scenarios but the model itself. The results of different OA components presented in Figure 3 were calculated as differences between two scenarios. For example, 'IVOC-SOA' is calculated as the difference between 'model_IVOC' and 'model_base'.